# Small Models are LLM Knowledge Triggers for Medical Tabular Prediction

**Jiahuan Yan**[1], **Jintai Chen**[2,3,*], **Chaowen Hu**[1], **Bo Zheng**[1], **Yaojun Hu**[1],
**Jimeng Sun**[3], **Jian Wu**[4,5]
[1]College of Computer Science and Technology, Zhejiang University
[2]Thrust of Artificial Intelligence, Information Hub, HKUST (GZ)
[3]Computer Science Department, University of Illinois Urbana-Champaign
[4]The Second Affiliated Hospital Zhejiang University School of Medicine
[5]Zhejiang Key Laboratory of Medical Imaging Artificial Intelligence
`{jyansir,chaowenhu,zjuzhengbo,yaojunhu,wujian2000}@zju.edu.cn,`
`jimeng@illinois.edu, jintaichen@hkust-gz.edu.cn`

## Abstract

Recent development in large language models (LLMs) has demonstrated impressive domain proficiency on unstructured textual or multi-modal tasks. However, despite with intrinsic world knowledge, their application on structured tabular data prediction still lags behind, primarily due to the numerical insensitivity and modality discrepancy that brings a gap between LLM reasoning and statistical tabular learning. Unlike textual or vision data (e.g., electronic clinical notes or medical imaging data), tabular data is often presented in heterogeneous numerical values (e.g., CBC reports). This ubiquitous data format requires intensive expert annotation, and its numerical nature limits LLMs' capability to effectively transfer untapped domain expertise. In this paper, we propose SERSAL, a general **self-prompting** method by synergy learning with small models to **enhance LLM tabular prediction in an unsupervised manner**. Specifically, SERSAL utilizes the LLM's prior outcomes as original soft noisy annotations, which are dynamically leveraged to teach a better small student model. Reversely, the outcomes from the trained small model are used to teach the LLM to further refine its real capability. This process can be repeatedly applied to gradually distill refined knowledge for continuous progress. Comprehensive experiments on widely used medical domain tabular datasets show that, without access to gold labels, applying SERSAL to OpenAI GPT reasoning process attains substantial improvement compared to linguistic prompting methods, which serves as an orthogonal direction for tabular LLM, and increasing prompting bonus is observed as more powerful LLMs appear. Codes are available at `https://github.com/jyansir/sersal`.

## 1 Introduction

The advancement of large language models (LLMs) (Zhao et al., 2023) has made waves in both research and industry communities. Through friendly natural language interaction and powerful in-context reasoning ability, LLMs have shown their remarkable knowledge generalization to language processing (Wei et al., 2021; Wang et al., 2022), complex planning (Qin et al., 2023; Zan et al., 2023) and even vertical domain (e.g., healthcare (Cascella et al., 2023), law (Deroy et al., 2023), chemistry (Guo et al., 2023)) tasks compared to past supervised pre-trained language models (Kenton & Toutanova, 2019; Radford et al., 2019), all achieved with suitable prompting and no fine-tuning, yet they are still struggling for the numeric tabular data.

For example, GPT-4 achieves 81.7 % accuracy with zero-shot prompting on the United States Medical Licensing Examination (USMLE), which metric will be increased to 90.2 % when meticulous prompts are designed (Nori et al., 2023). In the left part of Fig. 1(a), our preliminary experiment exhibits performances of GPT-3.5, GPT-4 and the fully supervised BERT on top-5 ICD coding for

---

*The corresponding author.

MIMIC-III discharge summaries. Even with simple zero-shot prompting, GPT-3.5 has already surpassed the fine-tuned ClinicalBERT (Huang et al., 2019) and can obtain further improvement with linguistic prompting tricks (e.g., zero-shot CoT (Kojima et al., 2022)). However, when handling medical tables with numerical value features, the trend becomes totally different, in the right part of Fig. 1(a), such significant prompting bonus disappears, suggesting an undeniable void in the current LLM prompting taxonomy tailored for tabular prediction. There are two key points causing the gap:

**(i)** Existing competitions for general-purpose LLMs predominantly focus on their capabilities in processing unstructured data (Zhang et al., 2024a;b), which is naturally different from structured tabular data characterized by dense heterogeneous numerical features (Borisov et al., 2022; Yan et al., 2023; Chen et al., 2022), and the prevailing technical landscape of LLMs neglects nuanced sensitivity and understanding for numerical values (Qian et al., 2023; Yan et al., 2024b).

**(ii)** Most LLM tasks of interest can be formulated as sample-level data understanding then reasoning by generation, the input semantics are unstructured and detailed, while the tabular prediction (e.g., disease diagnosis with numerical metrics from medical examination and testing) requires overall statistical relation between numerical features and targets on the whole dataset or a specific task, which is difficult to access using a single tabular instance in high-level and constrained contexts.

Based on these observations, a straightforward question is, how to harness world knowledge of existing versatile LLMs, especially these commercial and blackbox (users cannot access the logit) ones (OpenAI, 2022; 2023), to empower tabular prediction like disease diagnosis using medical testing results, which serves as a potential breakthrough for LLMs on statistical learning tasks.

To fill the aforementioned technical gap and extend LLM's capability to tabular prediction, we propose SERSAL, a **syner**gy learning pipeline between **s**mall models **a**nd **L**LMs, requiring no gold labels. Different from existing prompting techniques designing hard or soft prompts to augment inputs for unstructured data understanding, our SERSAL contributes from a brand new perspective that **improves existing LLMs on statistical prediction for numeric tabular data**, providing an interface to adapt LLM untapped knowledge to such structured tabular data. SERSAL helps a blackbox LLM trigger and refine its vertical capabilities for domain tabular data in an **unsupervised** manner with the following steps: (1) Using simple zero-shot prompting to gather the LLM's output confidence as initial coarse annotations of the whole dataset; (2) Teaching a better small tabular model (e.g., FT-Transformer (Gorishniy et al., 2021)) from scratch based on the LLM's confidence like semi-supervised learning with noisy labels; (3) Reversely fine-tuning the LLM using the outcomes of the trained small model to further update LLM's annotations in the next loop; The process from step (1) to (3) can be repeatedly applied to the LLM for continuous progress on specific tabular dataset. Essentially, SERSAL presents LLM prior knowledge on all tabular samples as "indicators" to a small model, which serves as a "probe" during learning correct patterns from the well-expressed clean part to feedback for LLM self-improvement.

In this paper, the main experiment is based on the well-known online blackbox LLMs, OpenAI GPT-3.5 (OpenAI, 2022) & GPT-4 (OpenAI, 2023), and as a prompting counterpart, our SERSAL can be directly transferred to other LLMs once the fine-tuning APIs are given. In a nutshell, our main contributions are:

- For the first time, we bring the common challenge of existing general-purpose LLMs on numeric tabular prediction, a statistical learning featured task, to the spotlight that has not been covered by prevailing prompting techniques.
- We propose SERSAL, a novel unsupervised self-prompting method to adapt LLM's capability to tabular data prediction, which leverages synergy learning with small models to capture and feedback correct patterns from LLM intrinsic knowledge.
- Comprehensive experiments reveal that SERSAL is consistently more effective than common textual prompting methods on medical tabular datasets, with general feasibility in other vertical domains discussed.

## 2 SERSAL: AN LLM SELF-PROMPTING LOOP FOR TABULAR PREDICTION

We propose SERSAL, a synergy learning process using small models to trigger LLM's knowledge on tabular data, which is a fundamentally distinct prompting method and serves as a novel inter-

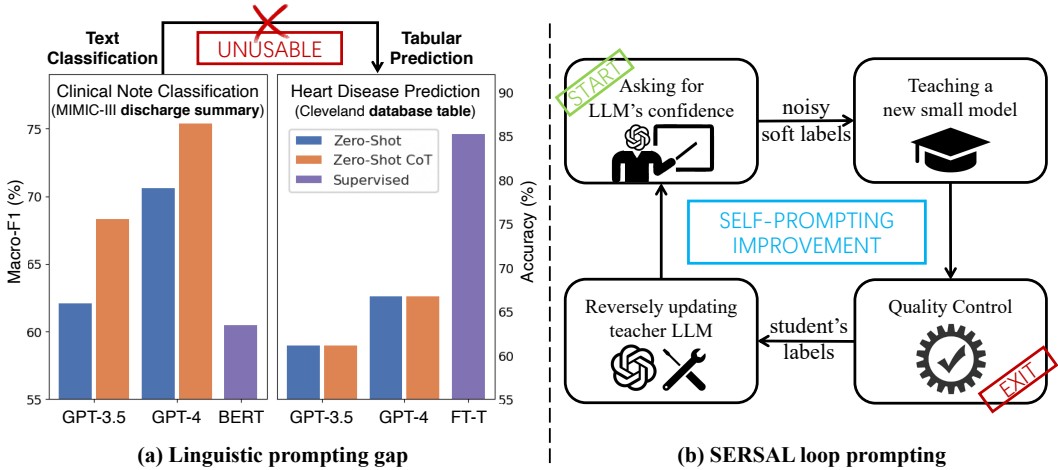

Figure 1: (a) Comparison of prompting effectiveness on unstructured textual data (Mullenbach et al., 2018) and structured tabular data (Detrano et al., 1989) from medical domain, it is clearly seen, even with surprising medical expertise (Nori et al., 2023), GPT-4 still struggles to catch up fully supervised small models (ClinicalBERT (Huang et al., 2019) for textual tasks and FT-Transformer (Gorishniy et al., 2021) for tabular ones) on tabular data, implying essential task discrepancy that makes it incompatible to rely on typical prompting techniques to unlock the potential of LLMs for tabular prediction. (b) Unsupervised SERSAL triggers LLM's knowledge using a small model.

face to extend current LLMs to tabular data prediction. Principally, SERSAL is inspired by the semi-supervised learning with noisy labels (LNL) and teacher-student training, while several key differences exist: (1) LNL setting requires a certain proportion of gold labels as the starting point, while SERSAL only access the LLM's **soft pseudo labels** (i.e., per-sample confidence) on the whole dataset; (2) In teacher-student paradigm the student model is primarily considered to be comparable to the teacher, while SERSAL conservatively **teaches a better student model** by dynamically learning from the relatively clean LLM's outputs and regularizing on noisy ones to avoid misleading confirmation bias (Tarvainen & Valpola, 2017), which produces a better small model on the target task to form a co-teaching manner. The overall framework of SERSAL is outlined in Fig. 1(b) and formulated in Algorithm 1. Each part is detailed in the following subsections.

## 2.1 SOFT LLM PSEUDO LABELING

To access the prior knowledge of the LLM on a specific tabular dataset, we first query its confidence on each sample using simple zero-shot prompt template. Specifically, the prompt consists of a task description and listed feature specifications, for example, "You are a professional doctor, here are some clinical metrics of a patient, please give a likelihood between 0 to 1 of suffering from a heart disease: [Age] 47 (years old); [Gender] Male; [Systolic Blood Pressure] 138 (mmHg); [Blood Lipid] 240 (mg/dL); . . .". In this way the LLM's per-sample confidence on the whole dataset is gathered, though the initial zero-shot performance is often far away from the one of a supervised small tabular model (see Fig. 1(a) and Table 2), we can dig into such fine-grained LLM confidence by separately judging then using underlying clean and noisy supervision signal to teach a robust small model.

## 2.2 SMALL MODEL TEACHING WITH NOISY LLM'S LABELS

This step aims to teach a better small model with the collected soft outputs from the LLM. Intuitively, such LLM confidence is a kind of noisy labels, thereby a straightforward insight is to reformulate the teaching process as learning with noisy labels (LNL). To sufficiently exploit LLM's prior knowledge, we adopt a "semi-supervised" learning strategy after dividing training samples into a more reliable labeled set and another unlabeled set, then the small model is fitted with the soft LLM's labels in the labeled set and regularized on the ones of the unlabeled set, the data partition is based on per-sample loss since deep neural networks tend to fit samples with clean labels faster than one with wrong

labels according to the LNL theory (Arpit et al., 2017), thus lower loss often indicates relatively cleaner labels (Chen et al., 2019).

In implementation, we use an adapted version of DivideMix (Li et al., 2019), a common semi-supervised LNL algorithm for image classification that dynamically fits a Gaussian Mixture Model on per-sample losses to distinguish probably clean and noisy LLM's labels and trains a pair of neural networks simultaneously to keep them diverged to avoid confirmation bias in single-model self-training (Tarvainen & Valpola, 2017). Apart from adapting DivideMix to tabular data prediction, the used soft labels naturally apply label smoothing guided by the LLM. Besides, we leverage the pseudo labels with extreme confidence for early stopping with underlying assumption that annotations with extreme LLM's confidence is tend to be more accurate, which is observed in Fig. 2 and Fig. 4, and discussed in Sec. 3.3. Specifically, we divide a training subset as the early stopping set $D_{es} = \{(\mathbf{X}_i, \bar{\mathbf{y}}_i)|\max(\hat{\mathbf{y}}_i) \geq \tau\}$ to perform early stopping and hyper-parameter selection for the teaching process, where for the $i$-th sample, $\hat{\mathbf{y}}_i$ is its LLM's confidence vector, and $\bar{\mathbf{y}}_i$ is the corresponding hard labels (i.e., $\bar{\mathbf{y}}_i = \mathrm{argmax}(\hat{\mathbf{y}}_i)$), samples with maximum label confidence larger than threshold $\tau$ (we uniformly set $\tau = 0.9$ in the experiment) are considered to be accurate enough for early stopping. During "semi-supervised" teaching, samples in the early stopping set are also used since some domain (e.g., medicine) tabular datasets suffer from data inadequacy, and the reduction on training subsets may distort data distribution. We formulate this step in the line 3-5 of Algorithm 1 and conduct related ablations in Sec. 3.3.

In summary, this teaching step adopts semi-supervised LNL process to aggregate and distill prior knowledge into a small model to extend the LLM's real capabilities to tabular data prediction.

## 2.3 QUALITY CONTROL

Since SERSAL operates iteratively, it requires a termination mechanism to control the loop exit. Here we provide three heuristic strategies, users can also define their own control flow in practice.

- **Metric-based Control.** In Sec. 2.2 we define the high-confidence training subset as the early stopping set $D_{es}$ which pseudo labels are relatively more accurate (see Fig. 2). Therefore, users can inspect metrics (e.g., AUC or accuracy scores for classification) by treating these pseudo labels as the "ground truth" to control whether to end the loop.

- **External Validation Control.** If budget permits, human experts can collect and annotate appropriate external data as a validation set, e.g., in hospitals, regular medical data quality inspection needs to sample and label a small part of data, and learning quality can be assessed with such external labeled set.

- **Rule-based Control.** For example, users can define a fixed iteration time.

For simplicity, in the main experiment we uniformly report one-loop SERSAL performances in medical and other domain datasets (Table 2 & 5), which has significantly surpassed the ones of prevailing prompting methods, and further discuss the effectiveness of multi-loop SERSAL in Sec. 3.4.

## 2.4 REVERSE LLM TUNING

The final step is to reversely teach the LLM using the well-trained small model to feedback the aggregated knowledge. Similar to using LLM's soft confidence to teach the small model in Sec. 2.2, we also use soft confidence from the small model to fine-tune the LLM (fine-tune the online black-box GPTs through their APIs in experiment). Specifically, the training samples are re-labeled by the small model with its guessed probabilities (line 7-8 in Algorithm 1), the same prompt templates in Sec. 2.1 are used to construct the training corpus for the LLM. To avoid the excessive memorization of the LLM on the small model outputs (Bordt et al., 2023), we employ a conservative tuning strategy that sets the maximum training epoch to 3 with proper early stopping (the fine-tuning APIs of GPT-3.5 & GPT-4 provide automatic early stopping in default), making the LLM slightly fitted on the guessed labels while keeping a non-zero minimum training loss. Then the updated LLM initiates the next SERSAL loop, forming an iterative process.

---

**Algorithm 1** Unsupervised `SERSAL`. Line 2: LLM pseudo labeling (Sec. 2.1); Line 3-5: Small model teaching (Sec. 2.2); Line 6: Quality control (Sec. 2.3); Line 7-9: Reverse tuning (Sec. 2.4).

---

**Input**: Unlabeled training set $\mathbf{X}_{\text{train}}$ and test set $\mathbf{X}_{\text{test}}$, large language model $f_{\text{LLM}}^{(0)}$
**Parameter**: Confidence threshold $\tau$, quality control function $f_{\text{ctr}}$
**Output**: Improved zero-shot tabular prediction $\mathbf{y}_{\text{test}}^*$
1: Let $t = 1$. // Initialize iteration number
2: Softly labeled dataset $D_{\text{train}}^{(t)} = (\mathbf{X}_{\text{train}}, \hat{\mathbf{y}}^{(t)})$ by current $f_{\text{LLM}}^{(t)}$.
3: Randomly initialize a small tabular model $\theta^{(t)}$.
4: Select early stopping set $D_{\text{es}}^{(t)} = \left\{ (\mathbf{X}_i, \bar{\mathbf{y}}_i^{(t)}) | \max(\hat{\mathbf{y}}_i^{(t)}) \geq \tau \right\} \subseteq D_{\text{train}}^{(t)}$.
5: $\theta^{*(t)} = \text{DivideMix}(D_{\text{train}}^{(t)}, D_{\text{es}}^{(t)}, \theta^{(t)}, \tau)$. // Adapted DivideMix (Li et al., 2019)
6: **while** $f_{\text{ctr}}(\theta^{*(t)}, \mathbf{X})$ **do**
7:     $\mathbf{y}_{\text{sm}}^{(t)} = \text{Predict}(\mathbf{X}_{\text{train}}; \theta^{*(t)})$. // Soft label guessing by the small model
8:     $\hat{\mathbf{y}}_{\text{sm}}^{(t)} = \text{Sharpen}(\mathbf{y}_{\text{sm}}^{(t)}, \text{temperature} = 0.1)$. // Simple temperature sharpening
9:     $f_{\text{LLM}}^{(t+1)} = \text{Finetune}(\mathbf{X}_{\text{train}}, \hat{\mathbf{y}}_{\text{sm}}^{(t)}, f_{\text{LLM}}^{(t)})$. // Reversely tune the LLM with guessed labels
10:     $t = t + 1$.
11:     Repeat Line 2-5. // Self-prompting loop
12: **end while**
13: $\mathbf{y}_{\text{test}}^* = \text{Predict}(\mathbf{X}_{\text{test}}; \theta^{*(t)})$. // Final prediction with the taught small model
14: **return** $\mathbf{y}_{\text{test}}^*$

---

## 3 EXPERIMENTS

In this section, we first compare `SERSAL` with prevailing prompting techniques (using GPT-3.5 & GPT-4) and the fully supervised small tabular models on extensive medical tabular datasets in Sec. 3.2. Next, we conduct ablation on several key adaptations in semi-supervised learning with noisy labels (LNL) in Sec. 2.2 and inspect the effectiveness of multi-loop `SERSAL` in Sec. 3.4. Also, we discuss the general adaptability of `SERSAL` on tabular data from other non-medical domains in Sec. 3.5. Besides, we explore the method interpretability by visualizing Shapely Value variation during `SERSAL` process in Sec. 3.6.

### 3.1 EXPERIMENTAL SETUP

**Datasets**    We evaluate on ten widely recognized medical diagnosis tabular datasets on various diseases: Heart Failure Prediction (HF, Detrano et al. (1989)), Lung Cancer Prediction (LC, Ahmad & Mayya (2020)), Early Classification of Diabetes (ECD, Islam et al. (2020)), Indian Liver Patient Records (LI, Ramana et al. (2012)), Hepatitis C Prediction (HE, Hoffmann et al. (2018)), Pima Indians Diabetes Database (PID, Smith et al. (1988)), Framingham Heart Study (FH, O'Donnell & Elosua (2008)), Stroke Prediction (ST, Fedesoriano (2020)), COVID-19 Presence(CO, Hemanthhari (2020)) and Anemia Disease (AN, Kilicarslan et al. (2021)). Besides, datasets in clinical trail (Wang & Sun, 2022) and open domains (Gorishniy et al., 2021) are added to further inspect the effectiveness of `SERSAL` in difficult tasks and general data domains respectively. We split each tabular dataset (80 % for training and 20 % for testing), and keep the same label distribution in each split. Statistics of medical diagnosis datasets are given in Table 1. All evaluated datasets are binary classification tasks.

| Dataset | HF | LC | ECD | LI | HE | PID | FH | ST | CO | AN |
|---|---|---|---|---|---|---|---|---|---|---|
| # features | 13 | 15 | 16 | 10 | 12 | 8 | 15 | 7 | 20 | 24 |
| # samples | 303 | 309 | 520 | 583 | 615 | 768 | 4238 | 5110 | 5434 | 15300 |
| P/N | 0.80 | 6.92 | 1.60 | 2.51 | 0.11 | 0.54 | 0.18 | 0.04 | 4.17 | 0.57 |
| disease | Heart | Lung | Diabetes | Liver | Hepatitis C | Diabetes | Heart | Stroke | COVID-19 | Anemia |

Table 1: Dataset statistics of ten medical diagnosis datasets for binary classification on various diseases. "P/N" denotes the amount ratio of positive samples and negative ones.

**Compared Methods** Since SERSAL serves as an unsupervised self-prompting method for LLM tabular prediction, we compare with existing linguistic prompting methods for LLM usage in general textual and tabular tasks, which focus on meticulously designed prompt texts: (1) **Zero-Shot Prompting** (0-shot) is the straightforward prompt that contains no examples; (2) **Zero-Shot CoT Prompting** (Kojima et al., 2022) (CoT) is a popular prompting method which asks the LLMs to answer with intermediate reasoning steps to enable complex reasoning capabilities; (3) **8-shot Prompting** (8-shot) is a common few-shot prompt setting in standard prompting studies (Wei et al., 2022; Kojima et al., 2022; Nori et al., 2023), it provides eight labeled samples (exemplars) to enrich prompt contexts and steer the LLM to the better outputs, in the experiment we randomly sample eight training examples and control the same positive-negative ratio (i.e., "P/N" in Table 1) with at least one example for each class; (4) **TabLLM** (Hegselmann et al., 2023) and (5) **LIFT** (Dinh et al., 2022) are two recent known linguistic prompt schemes for textualizing tabular data to fine-tune LLMs with gold labels, though TabLLM was additionally evaluated in zero-shot settings, **none of them are originally proposed for unsupervised tabular scenarios**, here we use their zero-shot schemes for comparison. Additionally, we provide a **fully supervised small tabular model (FSSM)** group using FT-Transformer (Gorishniy et al., 2021) for reference representing traditional supervised learning paradigm by fine-tuning dataset-specific small models.

**Implementation Details** All experiments are conducted with PyTorch on Python 3.8 and run on NVIDIA RTX 3090. For the small model, we uniformly use FT-Transformer with the default model and training configurations provided in the original paper (Gorishniy et al., 2021). For SERSAL, the only adjustable hyper-parameter is the temperature of DivideMix (Li et al., 2019) with choices of 0.5, 5.0 and 10.0 in line 5 of Algorithm 1, which is selected by the metric of the early stopping set ($D_{es}^{(t)}$ in line 4 of Algorithm 1). The LLMs in the experiment includes OpenAI GPT-3.5 & GPT-4 to inspect the effectiveness of SERSAL across different LLM capabilities.

## 3.2 WHY WE NEED SERSAL?

| | HF | LC | ECD | LI | HE | PID | FH | ST | CO | AN |
|---|---|---|---|---|---|---|---|---|---|---|
| Random guessing | 37.22 | 40.18 | 46.25 | 50.28 | 62.73 | 63.24 | 50.39 | 41.76 | 71.55 | 51.28 |
| FSSM*(supervised FT-T) | 88.19 | 86.61 | 99.60 | 78.94 | 100.00 | 84.72 | 66.25 | 82.98 | 99.91 | 99.92 |
| 0-shot (GPT-3.5) | 71.88 | 78.87 | 85.71 | 76.81 | 68.51 | 73.12 | 60.32 | 63.01 | 82.60 | 90.43 |
| 8-shot* (GPT-3.5) | 73.65 | 78.87 | **87.68** | 76.81 | 68.51 | 73.12 | 58.27 | 60.85 | 77.63 | 87.19 |
| CoT (GPT-3.5) | 71.88 | 78.87 | 82.36 | 76.81 | 68.51 | 70.83 | 60.32 | 63.01 | 82.60 | 90.43 |
| TabLLM (GPT-3.5) | 76.37 | 78.87 | 87.06 | 78.24 | 74.39 | 75.69 | 61.78 | 68.48 | 85.78 | 89.11 |
| LIFT (GPT-3.5) | 78.23 | 80.69 | 83.92 | 73.60 | 72.57 | 73.12 | 60.32 | 70.92 | 87.93 | 90.43 |
| SERSAL (GPT-3.5) | **91.39** | **85.42** | 86.40 | **79.39** | **85.14** | **78.97** | **63.97** | **76.36** | **96.85** | **98.37** |
| TabLLM+SERSAL (GPT-3.5) | 93.82 | 85.42 | 88.39 | 80.71 | 89.27 | 82.54 | 65.02 | 81.74 | 97.51 | 98.16 |
| SERSAL (GPT-4) | 94.18 | 86.93 | 92.68 | 82.51 | 92.76 | 82.39 | 67.14 | 81.23 | 97.96 | 98.82 |

Table 2: The AUC scores (%) of different tabular prediction schemes on 10 medical diagnosis datasets. The top part is the traditional supervised small models, the middle one is compared LLM prompting methods (the top performances are marked in **bold**), the bottom part is additional combinations. Here the results of SERSAL are only based on a single loop. "*" denotes the groups use gold labels. "FSSM" is the fully supervised FT-Transformer. The results on more difficult clinical trial datasets are given in Table 7.

**Main Results Analysis** The performances of different LLM prompting baselines are reported in the middle part of Table 2. An overall trend is that, when the GPT-3.5 meets medical domain tabular prediction tasks, the results using common prompting methods are consistently better than the ones of random guessing, demonstrating the general-purpose LLMs indeed contain medical domain expertise inherently, but they are still far from the traditional supervised small models (see group "FSSM"), and further performance enhancement can not be achieved through usual prompting tricks as in textual tasks (see Fig. 1(a)). Specifically, we observe 8-shot prompting slightly benefits the performances in small-scale datasets (e.g., HF and ECD) but hurts in the larger datasets (e.g., FH, ST, CO and AN) compared to the 0-shot prompting, which may be explained by the representativeness of the used examples, since the distribution of the smaller datasets are more likely to be covered by

few examples, thus 8-shot performs better as data scale decreases, and vice versa. For 0-shot CoT prompting, it does not affect the overall results in most cases, but we find slight performance decline in two diabetes datasets (i.e., ECD and PID), this may be caused by the over-consideration of CoT on noisy features since diabetes can be diagnosed with several prominent features (e.g., blood sugar and lipid). Although carefully crafted prompt templates from recent LLM in-context tabular learning studies (i.e., TabLLM and LIFT) show modest improvement, **they still follow the linguistic nature to process numeric tabular data**, and are primarily designed for LLM in-context few-shot learning or supervised fine-tuning. Our SERSAL **explores a fundamentally novel prompting mechanism exploiting the information gain in the LLM's noisy outputs**, which breaks through the predicament from an orthogonal perspective and serves as an interface to effectively adapt the LLM's domain knowledge to numeric tabular data. After applying SERSAL, without access to gold labels, the GPT-3.5 is able to achieve significantly better reasoning on these medical domain tasks, with many cases competitive with the supervised small models.

**Orthogonal Technical Contribution** Based on the above analysis, SERSAL works in a distinct underlying mechanism, and we can jointly adopt SERSAL and previous linguistic prompting methods for better combined performances (see group "TabLLM+SERSAL" in Table 2).

**Continuous Performance Growth** We additionally apply SERSAL to OpenAI GPT-4 on medical diagnosis datasets (the bottom part of Table 2) and more difficult clinical trial datasets (see Table 7). It can be seen SERSAL can further realize substantial performance gains as the capability of used LLMs becomes more powerful, which can even surpass the traditional supervised paradigm (N00041119 and N00312208 datasets in Table 7), indicating ample room for continuous prompting bonus in SERSAL alongside the emergence of more advanced LLMs.

## 3.3 Several Key Adaptations

|  | HF | LC | ECD | LI | HE | PID | FH | ST | CO | AN |
|---|---|---|---|---|---|---|---|---|---|---|
| SERSAL | 91.39 | 85.42 | 86.40 | 79.39 | 85.14 | 78.97 | 63.97 | 76.36 | 96.85 | 98.37 |
| w/o soft pseudo | 84.58 | 76.58 | 87.24 | 78.25 | 75.79 | 75.93 | 62.58 | 75.05 | 93.97 | 97.53 |
| w/o ES | 84.03 | 74.11 | 75.92 | 59.39 | 47.41 | 68.43 | 57.08 | 74.70 | 90.57 | 97.57 |

Table 3: The AUC scores of ablation on two key adaptations. "w/o soft pseudo" means replacing the LLM's soft outputs with hard ones during teaching the student model, "w/o ES" denotes no early stopping during DivideMix (line 5 in Algorithm 1).

In Sec. 2.2, to adapt the LLM's outputs to a semi-supervised LNL process to teach a small model, we gather per-sample confidence from the LLM as soft noisy annotations and heuristically select samples with extreme confidence for early stopping. In this section, we will analyze the effect of the two designs which distinguish our SERSAL from traditional LNL settings.

**The Effect of using Soft Labels** We query soft confidence from the LLM (see Sec. 2.1) rather than directly using hard outputs for small model teaching. The prediction probabilities inherently reflect the LLM's prior knowledge as well as uncertainty on the domain tabular data and can be naturally treated as a kind of label smoothing. Besides, the probability values can be used to select relatively reliable labels to early stop the teaching process and avoid over-fitting. In Table 3 we compare the effect of using soft labels by replacing it with hard ones during

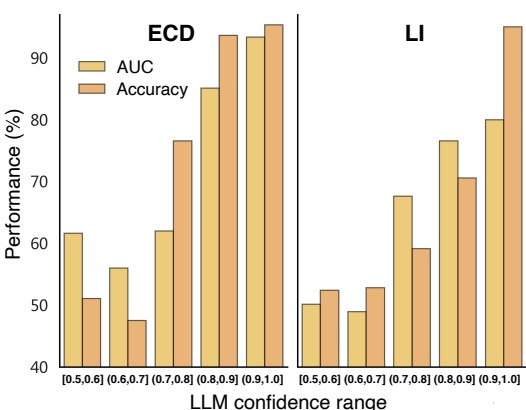

Figure 2: Performances in different LLM's confidence ranges on ECD and LI datasets. Extreme-confidence samples are relatively more reliable.

SERSAL reasoning (group "w/o soft pseudo"). We find that using hard ones is usually suboptimal since it loses both prediction uncertainty and label smoothing, which is unable to exploit fine-grained LLM's knowledge.

**The Effect of Early Stopping** In addition to using LLM's soft outputs, a relatively clean training subset is selected by threshold clipping on the per-sample confidence (line 4 in Algorithm 1) for early stopping. Table 3 report the ablation results by directly training 100 epochs (group "w/o ES"). It can be clearly seen, simply following the original DivideMix is far from the desired results, since tabular features are heterogeneous and high-level compared to the well-patterned pixels of images (Chen et al., 2023; Yan et al., 2024b), and in medical tabular domain the typically limited available data further makes it prone to overfit without early stopping, for example, except large AN dataset, all other tabular datasets appear to be significantly impacted by removing the early stopping mechanism. The heuristic design of selecting extreme-confidence sample is inspired from the empirical assumption that confident predictions from the LLM are more likely to be accurate, which is supported by the performance variation of different confidence ranges in Fig. 2 and Fig. 4.

### 3.4 EFFECTIVENESS OF MULTI-LOOP SERSAL

Since SERSAL can be iteratively applied to the LLM (see Fig. 1(b)), we further inspect the effectiveness of multi-loop SERSAL for GPT-3.5 reasoning. Specifically, we repeat the pipeline three times on ECD and LI datasets, the result variations are reported in Table 4.

During three loops, progressive improvement on both the small model (SERSAL outputs are from the well-trained small model of each loop) and the GPT-3.5 is observed. Surprisingly, even inferior to the 8-shot prompting baseline on ECD dataset after the first loop (see Table 2), we find SERSAL can reduce the gap and even outperform few-shot baselines after several loops. Such continuous progress probably comes from the synergy learning between the small model and the LLM that **shares a similar underlying mechanism of co-teaching** (Han et al., 2018), i.e., both sides dynamically learn from a part of reliable pseudo labels from each other and it makes them diverged to avoid confirmation bias, forming a mutual improvement manner to aggregate and refine LLM's untapped domain knowledge for tabular prediction.

| # Loop | ECD | | LI | |
|---|---|---|---|---|
| | SERSAL | LLM 0-shot | SERSAL | LLM 0-shot |
| 1 | 86.40 | 85.71 | 79.39 | 76.81 |
| 2 | 87.00 | 86.42 | 82.47 | 80.26 |
| 3 | 89.00 | 87.81 | 84.07 | 82.91 |

Table 4: The AUC score variation of SERSAL outputs and zero-shot prompting of the tuned GPT-3.5 (LLM 0-shot) on LI and ECD datasets during three loops. "# Loop" is the same as the variable $t$ in line 1 of of Algorithm 1. LLM 0-shot group at the first loop is the original LLM.

### 3.5 GNERALIZED DATA ADAPTABILITY ON OTHER DOMAINS

In this section, we further explore the data adaptability of SERSAL on other non-medical domains. We evaluate on three classic binary classification datasets: Churn Modeling (Iyyer, 2019), Credit (Credit Fusion, 2011) and Adult (Kohavi et al., 1996), which are widely included in general tabular prediction studies (Gorishniy et al., 2021; Yan et al., 2023; Grinsztajn et al., 2022). Additionally, we build a dataset "Fake" by randomly generating features and binary labels to emulate an extreme case where the LLM has no relevant knowledge at all. The data information and the results are given in Table 5. As in the medical domain, the GPT-3.5 indeed holds the world knowledge and can directly achieve the considerable results with simple zero-shot prompting, and SERSAL further enhances the performance significantly. However, when facing the tabular data from an unknown

| | Churn | Credit | Adult | Fake |
|---|---|---|---|---|
| domain | Business | Finance | Sociology | N/A |
| # features | 10 | 10 | 14 | 6 |
| # samples | 10000 | 16714 | 48842 | 1000 |
| Random guessing | 66.35 | 43.80 | 58.73 | 53.85 |
| FSSM* | 86.27 | 84.88 | 91.39 | 55.31 |
| 0-shot (GPT-3.5) | 77.81 | 69.05 | 75.10 | 46.28 |
| SERSAL (GPT-3.5) | 83.29 | 79.36 | 88.72 | 38.72 |

Table 5: The dataset statistics and AUC scores on other non-medical domains. "Fake" is a generated dataset with random labels and features. The denotations follow the ones in Table 1 and Table 2.

domain (i.e., the Fake dataset), the LLM outputs high confidence on wrong labels, `SERSAL` is unable to recognize such totally misleading bias. Therefore, our `SERSAL` shares the same basic limitation as other linguistic prompting methods that the applied LLMs require a certain level of knowledge in the target domain.

## 3.6 INTERPRETABILITY OF SERSAL

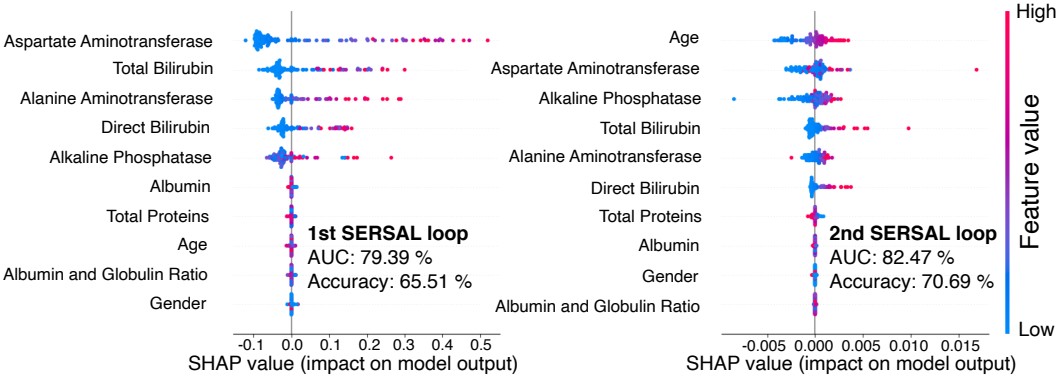

Figure 3: Interpretability visualization from feature importance perspective: the variation of the Shapley Values (treat `SERSAL` outputs as the targets) and performances on Indian Liver Patient Records (LI dataset) after one and two `SERSAL` loops using GPT-3.5.

In Fig. 3 we visualize the variation of Shapely Values on Indian Liver Patient Records (LI) dataset after one (left) and two (right) `SERSAL` loops by treating the predictions (i.e., Algorithm 1) as targets. It can be clearly seen the feature "Age" is adequately considered after one loop self-prompting, which highlights a strong and reasonable positive correlation between age and the incidence of liver diseases that aligns with the medical expertise. Besides, a negative correlation with "Total Proteins", a guiding clinical metric reflecting the liver's synthetic function, is captured in the right figure to contribute the prediction, since a lower total protein level indicates a risk of liver cirrhosis. These two reasonable changes of feature importance interpret the `SERSAL` prompting is able to iteratively refine the domain expertise in the LLM, calibrating the statistical feature-target relationship for better reasoning results during the process.

## 4 RELATED WORK

**Prompt Engineering for In-Context Learning** Prompt engineering is a flourishing discipline for better LLM reasoning through meticulously designed linguistic input contexts or interaction process. The most common and straightforward prompting is the single-round instruction that directly asks with zero (zero-shot) or several (few-shot) demonstrations (Brown et al., 2020; Wei et al., 2021), but such prompt style fails to work in more complex reasoning tasks (Wei et al., 2022). To tackle this deficiency and improve the LLM's capacity on a wide range of tasks, recently, studies on more advanced prompting methods are emerging, such as chain-of-thought (CoT) (Wei et al., 2022; Kojima et al., 2022; Zhang et al., 2023a), tree-of-thought (ToT) (Yao et al., 2023) and self-consistency (Wang et al., 2023). However, current prompting methods are mostly designed to serve unstructured data tasks (Zhang et al., 2023b). Although recent studies on LLM in-context learning for tabular data (e.g., TabLLM (Hegselmann et al., 2023), LIFT (Dinh et al., 2022)) propose table-friendly prompting strategies, their linguistic nature still hinders the numeric table understanding (Yan et al., 2024b).

**Semi-supervised Learning with Noisy Labels** Semi-supervised learning treats the unlabeled samples as regularization for better model generalization (Lee et al., 2013; Tarvainen & Valpola, 2017; Miyato et al., 2019; Berthelot et al., 2019). Recently, the related theory has been introduced to noisy label learning scenarios (Song et al., 2022) that dynamically divide samples into clean labeled group and noisy unlabeled one (Li et al., 2019) to achieve robust learning from noisy labels.

**LLMs for Tabular Data Prediction**    As a machine learning task in tabular data applications, tabular prediction has gained increasing attention from the research community due to the heterogeneous nature and numerical features of tabular data compared to other unstructured modalities. Previous studies in tabular prediction models focus on designing tailored neural networks (Arik & Pfister, 2021; Gorishniy et al., 2021; Yan et al., 2024a; Chen et al., 2024) to emulate and surpass traditional tree-based models (Chen & Guestrin, 2016; Ke et al., 2017) under fully supervised paradigm. More recently, motivated by the widespread success of pre-trained language models (Brown et al., 2020; OpenAI, 2022; 2023), the unique bonus of neural networks is exploited in tabular model development, such as pre-training (Wang & Sun, 2022) and in-context learning Hollmann et al. (2023), and open-sourced LLMs have been popular base models to be adaptively pre-trained for better tabular prediction since their inherent knowledge (Yan et al., 2024b; Wen et al., 2024). However, current exploration on tabular prediction LLMs involves costly pre-training on large-scale tabular data, which requires access to LLM codes and parameters, and heavy adaptation to tabular data may impact the original usability of the LLMs on other unstructured data tasks.

## 5    CONCLUSIONS

This paper revealed the common challenge of existing general-purpose LLMs on tabular prediction and proposed SERSAL, a novel unsupervised self-prompting method in non-linguistic mechanism that triggers the LLM's domain knowledge for better tabular prediction. This is achieved through a co-teaching process between the LLM and a well-taught small tabular model which learn from the other's noisy outputs to aggregate and refine the LLM's untapped capabilities. Extensive experiments on medical and non-medical domain tabular datasets demonstrate that, as an orthogonal prompting landscape, SERSAL is consistently suitable for extending the potential of LLMs to numeric tabular data.

### ACKNOWLEDGMENTS

This research was partially supported by National Natural Science Foundation of China under grant No. 12326612, Zhejiang Key R&D Program of China under grant No. 2023C03053, and Zhejiang Key Laboratory of Medical Imaging Artificial Intelligence.

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

## A  LIMITATIONS & IMPACTS

As discussed in Sec. 3.5, though our SERSAL is distinguished from traditional prompting methods by its non-linguistic mechanism, it still requires the LLMs with latent knowledge in the target domain to be effective. Therefore, in practice the user should have prior understanding of the used LLM's capability or advantageous application fields. SERSAL contributes to the progress in both LLM prompting and tabular data community through providing a novel interface to adapt untapped knowledge in LLMs to the tabular prediction tasks in a zero-shot manner, which is particularly useful in the regime where limited data or annotation is available.

## B  DATASETS AND EXPERIMENT DETAILS

We provide detailed data information of the experiment tabular datasets in Table 6. We drop the samples with missing features and adopt the same preprocessing as Gorishniy et al. (2021) before training. For MIMIC-III discharge summary dataset (Johnson et al., 2016; Mullenbach et al., 2018) used in Fig. 1(a), we retain the most frequent 5 labels (medical codes) since our goal is just to demonstrate the prompting effectiveness on medical textual tasks and conducting validation on the full label version (several thousands labels) is inconvenient. During conducting zero-shot prompting for GPT-3.5v and GPT-4v on the MIMIC-III dataset, we follow the PhysioNet Credentialed Data Use Agreement [1] and enroll in the Azure OpenAI service without human review of the data to protect the data from third-party access.

| Dataset | Abbr. | # Sample | # Feature | P/N | Source Link |
|---|---|---|---|---|---|
| Indian Liver Patient Records | LI | 583 | 10 | 2.51 | https://www.kaggle.com/datasets/uciml/indian-liver-patient-records |
| Pima Indians Diabetes Database | PID | 768 | 8 | 0.54 | https://www.kaggle.com/datasets/uciml/pima-indians-diabetes-database |
| Framingham Heart Study | FH | 4238 | 15 | 0.18 | https://www.kaggle.com/datasets/mohannapd/ramingham-heart-study |
| Stroke Prediction | ST | 5110 | 7 | 0.04 | https://www.kaggle.com/datasets/fedesoriano/stroke-prediction-dataset |
| Hepatitis C Prediction | HE | 615 | 12 | 0.11 | https://www.kaggle.com/datasets/fedesoriano/hepatitis-c-dataset |
| COVID-19 | CO | 5434 | 20 | 4.17 | https://www.kaggle.com/datasets/hemanthhari/symptoms-and-covid-presence |
| Lung Cancer Prediction | LC | 309 | 15 | 6.92 | https://www.kaggle.com/datasets/mysarahmadbhat/lung-cancer |
| Heart Failure Prediction | HF | 303 | 13 | 0.80 | https://archive.ics.uci.edu/dataset/45/heart+disease |
| Early Classification of Diabetes | ECD | 520 | 16 | 1.60 | https://www.kaggle.com/datasets/andrewmvd/early-diabetes-classification |
| Anemia Disease | AN | 15300 | 24 | 0.57 | https://www.kaggle.com/datasets/serhathoca/anemia-disease |
| Churn Modeling | - | 10000 | 10 | 0.26 | - |
| Give Me Some Credit | - | 16714 | 10 | 1.00 | https://www.kaggle.com/c/GiveMeSomeCredit |
| US Adult Income | - | 48842 | 14 | 0.31 | https://www.kaggle.com/datasets/johnolafenwa/us-census-data |

Table 6: Detailed data information of used tabular datasets (10 from the medical domain and 3 from others). "P/N" denotes the amount ratio of positive samples and negative ones.

## C  RESULTS ON CLINICAL TRIAL DATASETS

We evaluate SERSAL on clinical trail mortality datasets, which require specialized scientific knowledge for clinical trials. Although SERSAL prompting with GPT-3.5 cannot directly achieve good performance on such vertical tasks, further performance gains are still observed once we use more powerful GPT-4, indicating room for continuous improvement as more advanced LLMs appear.

| | N00041119 | N00174655 | N00312208 | N00079274 | N00694382 |
|---|---|---|---|---|---|
| FSSM*(supervised FT-T) | 62.38 | 89.20 | 77.83 | 71.78 | 73.89 |
| 0-shot (GPT-3.5) | 56.79 | 73.08 | 63.49 | 59.85 | 62.70 |
| CoT (GPT-3.5) | 56.79 | 73.08 | 60.73 | 59.85 | 62.70 |
| SERSAL (GPT-3.5) | **58.31** | **82.64** | **71.92** | **64.17** | **66.31** |
| SERSAL (GPT-4) | 65.08 | 88.62 | 78.39 | 67.94 | 71.47 |

Table 7: The AUC scores (%) of different tabular prediction schemes on clinical trail mortality datasets used in Wang & Sun (2022) (see ClinicalTrials.gov). The similar denotations are used as Table 2. No gold labels are used for prompting methods here. It can be seen SERSAL can achieve continuous improvement and even perform comparably with the traditional supervised paradigm once more powerful base LLMs are applied.

---

[1] https://physionet.org/news/post/415

# D   MECHANISM EXPLANATION OF DIVIDEMIX IN SERSAL

To make the paper friendly to the audiences from different background, in this section we provide detailed mechanism explanation of learning with noisy labels (LNL) and how to learn a better small (neural network) model from LLM noisy annotations using DivideMix.

**DivideMix mechanism in SERSAL**   In the traditional noisy data learning field, it was theoretically proved and empirically observed that the "memorization" behavior of neural networks leads to different optimization behavior on real data and noisy ones that neural networks tend to learn simple patterns first before fitting label noise (Arpit et al., 2017). Based on this theoretical foundation, a typical group of LNL methods (Berthelot et al., 2019; Li et al., 2019) exploit per-sample training loss to judge the noisy labels, for example, in our paper we adopt DivideMix (Li et al., 2019) to learn a small model using LLM noisy annotations, which models the noise probabilities of each sample by dynamically fitting a Gaussian Mixture Model (GMM) on per-sample losses, all training samples are divided into a clean set and a noisy set based on a probability threshold $\tau$. During the DivideMix training process, samples in the clean set are used for supervised learning (using their soft LLM annotations), while ones in the noisy set is used in an unsupervised manner (only using their features), e.g., learn with regularization loss or reconstruction task. The process will be ended until the average loss of heuristically selected early stopping subset (high-LLM-confidence samples $D_{es}$ in Algorithm 1) is converged, i.e., the loss of early stopping subset is not decreased for $m$ epochs. Notably, clean sample is not equivalent to high-LLM-confidence sample, but the sample which LLM annotation is easier to fit by the small tabular model. Since the small model (i.e., FT-Transformer here) is only supervised by clean data and regularized on noisy data, all data is sufficiently and reasonably exploited to acquire a better pattern.

**DivideMix hyperparameters in SERSAL**   We refer to the original hyperparameter settings in DivideMix paper [4] and only search the temperature ($T$) in $\{0.5, 5.0, 10.0\}$, with fixed regularization loss weight $L_u$ to 25, clean probability $\tau$ to 0.9, and the learning rate of the small model (FT-Transformer) to 1e-4. Additionally, we uniformly introduce the early stopping patience $m$ to 5. The best temperature is selected based on the training loss of early stopping subset $D_{es}$.

|                  | HF    | LC    | ECD   | LI    | HE    | PID   | FH    | ST    | CO    | AN    |
|------------------|-------|-------|-------|-------|-------|-------|-------|-------|-------|-------|
| 0-shot GPT-3.5 #1 | 71.88 | 78.87 | 85.71 | 76.81 | 68.51 | 73.12 | 60.32 | 63.01 | 82.60 | 90.43 |
| SERSAL #1         | 91.39 | 85.42 | 86.40 | 79.39 | 85.14 | 78.97 | 63.97 | 76.36 | 96.85 | 98.37 |
| 0-shot GPT-3.5 #2 | 87.58 | 83.74 | 86.42 | 80.26 | 86.18 | 79.26 | 63.86 | 73.62 | 91.29 | 93.62 |
| SERSAL #2         | 92.03 | 86.15 | 87.00 | 82.47 | 87.32 | 80.61 | 65.27 | 79.58 | 97.20 | 98.93 |
| 0-shot GPT-3.5 #3 | 89.26 | 85.39 | 87.81 | 82.91 | 86.87 | 81.47 | 64.12 | 76.37 | 93.65 | 94.13 |
| SERSAL #3         | 93.58 | 85.42 | 89.00 | 84.07 | 89.57 | 81.83 | 65.27 | 80.93 | 97.02 | 98.60 |

Table 8: The AUC score variation of `SERSAL` outputs and zero-shot prompting of the tuned GPT-3.5 on all datasets from Table 2 during three loops.

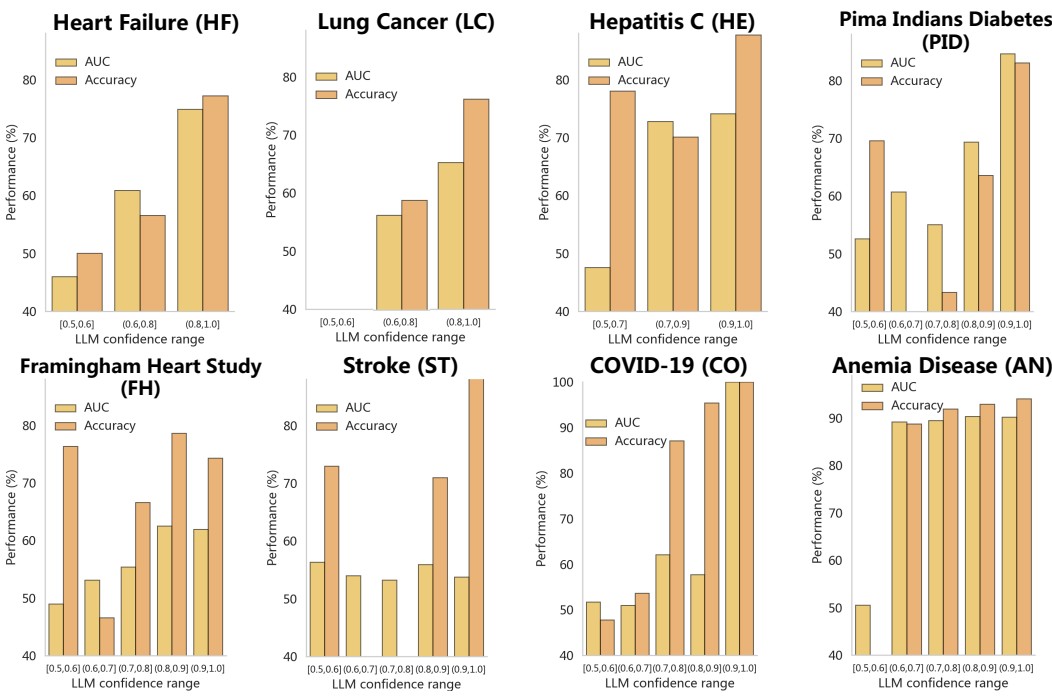

Figure 4: Performances in different LLM confidence ranges on other eight datasets. The overall trend of high-confidence samples being relatively more reliable still holds.

