# OpenReview forum: "Small Models are LLM Knowledge Triggers for Medical Tabular Prediction"
_ICLR.cc/2025/Conference — ICLR 2025 Poster_

### Official Review · Reviewer_uFdg · 2024-10-30

**Soundness:** 3
**Presentation:** 2
**Contribution:** 3
**Rating:** 5
**Confidence:** 4

**Summary:**

The authors propose SERSAL, a general loop of thought prompting method by synergy learning with small models to unconditionally enhance tabular prediction for LLMs in an unsupervised way. This paper mainly solves the medical tabular prediction tasks.

**Strengths:**

1. The direction of applying synergy learning with small models and LLMs to enhance tabular prediction in an unsupervised way is promising and important.
2. The paper is easy to follow.
3. The proposed method is novel.

**Weaknesses:**

Please refer to questions below.

**Questions:**

1. Does the paper mainly to solve traditional tabular prediction tasks or medical prediction tasks? If it's former, it seems that the used tabular data does not contain regression and multi-class classification, as discussed in [1]. So how does it behave on traditional tabular data? If it's latter, how do the authors ensure the safety of processing medical data with Large Language Models (LLMs), especially when adapting to real-world healthcare scenarios where data security is crucial [2], and the data is used for fine-tuning the LLMs? In addition, while the paper seems to position medical tabular prediction tasks as the central focus, the proposed method appears to be more suited for general tabular learning rather than specialized medical prediction. This perception arises from the lack of incorporation of specific medical domain knowledge within the paper. Could the authors give more explanations?
[1] Revisiting Deep Learning Models for Tabular Data. NeurIPS 2021.
[2] Large language models encode clinical knowledge. Nature 2023.

2. I am uncertain whether it would be more appropriate to refer to the setting as unsupervised rather than zero-shot. In the context of zero-shot learning, as illustrated in TabLLM [3] and GTL [4], zero-shot implies that in target dataset, no training instances (x) that are similar to test instances are provided. In this paper, in target dataset, training instances from the same distribution without labels are provided for training. What is the definition of zero-shot used in this paper? Does 'no label' equate to 'zero-shot'?
[3] TabLLM: Few-shot Classification of Tabular Data with Large Language Models.
[4] From Supervised to Generative: A Novel Paradigm for Tabular Deep Learning with Large Language Models.


3. Is the annotation of samples by LLMs done on a sample-by-sample basis? If so, this process seems to require a high cost. Could you provide the time cost?

4. Is TabLLM in Table 2 be tested in a zero-shot scenario? If so, it seems that TabLLM is better than GPT-3.5. Could you please explain why this might be the situation, given that GPT 3.5 has a more extensive knowledge base compared to TabLLM?

5. What is the meaning of TabLLM + SERSAL (GPT-3.5) in Table 2, what is the role of each component?

6. Recently, generative models pertained on a large set of tabular data have also demonstrated zero-shot capabilities on tabular prediction tasks [4]. I suggest that these models should be considered as baseline models since they can be seen as powerful tools for zero-shot performance on new tasks.
[4] From Supervised to Generative: A Novel Paradigm for Tabular Deep Learning with Large Language Models.

---

> ### Author Response · Authors · 2024-11-24
> **To uFdg (1/4, Q1)**
>
> We sincerely appreciate your insightful questions & constructive suggestions that help us deeply improve the paper, and finding our work promising, important, novel and follower-friendly.
>
> ---
>
> **Q1:** Does the paper mainly to solve traditional tabular prediction tasks or medical prediction tasks?
>
> **A1:** Thank you for your important question to make a clearer positioning of our paper more. We would like to answer in two steps.
>
> (1) Focus on traditional tabular prediction or just medical domain tabular task?
>
> We focus on traditional tabular prediction.
>
> (2) Is there more cases in diverse data domains (non-medical) & task types (i.e., multi-class, regression tasks?
>
> (2-a) According to the suggestion, we provide additional evaluation on your mentioned benchmark from [1] (including binary, multi-class classification and regression datasets), we only consider the muti-class datasets with class number smaller than 10 (since too many class number has no concrete meaning for each class and exceed the maximum context length of GPT-3.5), and for the datasets with more than 100K samples, we uniformly sub-sample it to 100K for saving the cost of ChatGPT APIs, the results are based on the single-loop SERSAL.
>
> |                           |     CA     |    AD    |      JA     |       YE      |       CO      |
> |---------------------------|:----------:|:--------:|:-----------:|:-------------:|:-------------:|
> | task type                 | regression | binclass | multi-class |    binclass   |  multi-class  |
> | data scale ($N$)            |    20640   |   48842  |    83733    | 100K(sampled) | 100K(sampled) |
> | feature scale ($F$)         |      8     |    14    |      54     |       90      |       54      |
> | class number              |     n/a    |     2    |      4      |      n/a      |       7       |
> | supervised FT-Transformer |    0.459   |   91.39  |    73.26    |     9.238     |     92.63     |
> | 0-shot prompting(GPT-3.5) |    0.538   |   75.10  |    48.96    |     11.536    |     62.39     |
> | SERSAL(0-shot, GPT-3.5)           |    0.496   |   88.72  |    62.37    |     10.285    |     78.72     |
>
> Adult dataset in table 5 is from this benchmark as well. Originally, we did not include multi-class tasks since they can be decomposed into several binary classification tasks; While for regression task it is another story, since our adapted noisy data learning method (i.e., DivideMix [2]) is inspired from noisy image learning, in which field the tasks are all benchmarked on the noisy classification, and noisy regression task has not been explored yet; but in principle, these methods just rely on per-sample training loss to judge the noisy labels. Therefore, for the first time we try to explore the noisy label learning on regression task here, and the improvement of using SERSAL for tabular regression is also obvious. We will curate these results into the final paper version to deepen the general adaptability of data domain and task type of our SERSAL, and make the paper positioning clearer. Thank you for the advice so much!
>
> (2-b) Although SERSAL does not focus on medical tasks, we still briefly discuss its possible usage in medical practice. Prediction safety & data safety are two major concerns in medical applications, for prediction safety we think it is more suitable for SERSAL to be an auxiliary method to vote for a real diagnosis together with human experts, and the SERSAL results will be more reliable as more powerful LLMs appears (see Table 7); For data safety, it is more practical to use private foundational LLMs / domain models continuously pre-trained on private clinical corpus and fine-tuned on the patient data during usage.
>
> (2-c) As mentioned in the Weakness 1&Q1 in our reply for the third Reviewer jQZe (we provide results of SERSAL on multi-label & multi-class clinical text datasets there), the entire value for SERSAL is not only on tabular data modality, it is also usable for textual or other non-tabular data, we propose SERSAL and emphasize its usage in tabular data in this paper considering the fact that traditional textual (linguistic) prompting techniques are ineffective for numerical tabular data (LLM numerical insensitivity [3]) in contrast to their success in unstructured data in the LLM community.
>
> **Reference:**
>
> [1] Revisiting Deep Learning Models for Tabular Data. NeurIPS 2021.
>
> [2] DivideMix: Learning with noisy labels as semi-supervised learning, ICLR 2019.
>
> [3] Making Pre-trained Language Models Great on Tabular Prediction, ICLR 2024.

---

> ### Author Response · Authors · 2024-11-24
> **To uFdg (2/4, Q2&Q3)**
>
> **Q2:** I am uncertain whether it would be more appropriate to refer to the setting as unsupervised rather than zero-shot. In the context of zero-shot learning, as illustrated in TabLLM and GTL, zero-shot implies that in target dataset, no training instances ($X$) that are similar to test instances are provided. In this paper, in target dataset, training instances from the same distribution without labels are provided for training. What is the definition of zero-shot used in this paper? Does 'no label' equate to 'zero-shot'?
>
> **A2:** We appreciate your indeed careful and rigorous considerations, and yes, the definition of ‘zero-shot’ in this paper is equivalent to ‘no (gold) label’, and ‘zero-shot tabular prediction’ here is ‘tabular prediction without (gold) labels’. Technically speaking, zero-shot learning refers to the scenario where the target-domain training data is inaccessible, thus it is indeed more suitable to treat the case of SERSAL using unlabeled target-domain data here as unsupervised learning. The expression ‘zero-shot prediction’ is inspired from ‘zero-shot prompting’ since these textual prompting methods are used as baselines. We are going to emphasize the unsupervised nature of SERSAL in keywords, abstract and the method description to address the confusion in the final paper version.
>
> ---
>
> **Q3:** Is the annotation of samples by LLMs done on a sample-by-sample basis? If so, this process seems to require a high cost. Could you provide the time cost?
>
> **A3:** For small datasets ($N < 1000$), we use per-sample LLM inference for sample annotation, while for large ones ($N \ge 1000$), multiple ($k$) samples are stacked in one query to save time, the sample amount in each query is decided by the maximum context length (e.g., 4096 tokens for GPT-3.5) of the base LLM, we list the annotation time of the datasets in Table 2 as follows, and we add inference time to show its great online inference efficiency.
>
> | Dataset                                      |  HF  |  LC  |  ECD |   LI  |   HE  |  PID  |  FH  |  ST  |  CO  |   AN  |
> |----------------------------------------------|:----:|:----:|:----:|:-----:|:-----:|:-----:|:----:|:----:|:----:|:-----:|
> | data scale ($N$)                               |  303 |  309 |  520 |  583  |  615  |  768  | 4238 | 5110 | 5434 | 15300 |
> | feature scale ($F$)                            |  13  |  15  |  16  |   10  |   12  |   8   |  15  |   7  |  20  |   24  |
> | sample amount per query ($k$)                  |   1  |   1  |   1  |   1   |   1   |   1   |  30  |  60  |  30  |   30  |
> | stacked query (yes or no)                    |  no  |  no  |  no  |   no  |   no  |   no  |  yes |  yes |  yes |  yes  |
> | GPT-3.5 annotation time all (second)         | 697  | 587  | 780  | 1516  | 1169  | 1152  | 398  | 275  | 546  | 1632  |
> | GPT-3.5 annotation time per query   (second) | 2.3  | 1.9  | 1.5  |  2.6  |  1.9  |  1.5  | 2.8  | 3.2  | 3.0  |  3.2  |
> | SERSAL inference time per sample (second)    | 0.03 | 0.03 | 0.03 |  0.03 |  0.03 |  0.02 | 0.03 | 0.02 | 0.03 |  0.04 |
>
> Since SERSAL supports powerful online blackbox LLMs and we experiment on ChatGPT (GPT-3.5 or 4), the annotation time is majorly affected by real-time network latency and underlying optimization engineering of ChatGPT APIs rather than context length. Therefore, SERSAL algorithm is positioned at offline training process and not designed for online / real-time training, and we can further save more time by stacking more samples into one query for the small datasets here. Besides, in Table 2, 1-loop SERSAL takes the same annotation time as other textual prompting baselines but already has acquired significantly superior performances. Also, once SERSAL training phase is finished, its inference (using the small model) cost is small enough for online service.

---

> > ### Comment · Reviewer_uFdg · 2024-11-26
> >
> > Many thanks to authors. If the authors agree that it would be more appropriate to refer to the setting as unsupervised rather than zero-shot, I am uncertain whether the title and main description about zero-shot in the paper should be changed?

---

> > > ### Author Response · Authors · 2024-11-26
> > >
> > > Of course, we agree with your opinion that our SERSAL is more appropriate to be treated as an unsupervised data setting rather than zero-shot one, since the future audiences may treat the "zero-shot tabular prediction" as "zero-shot learning", causing potential confusions as your mentioned question Q2, what we want to convey in fact is no gold labels. We greatly value your advice and plan to revise the title to "Synergy Learning with Small Models promotes LLM Unsupervised Tabular Prediction” with all the corresponding expressions changed in the main description in the final paper version to avoid potential misunderstanding! At present, we cannot change our title, which can only be revised during camera ready phase. Thank you for improving our paper for a clearer positioning and making it reader-friendly!

---

> > > ### Author Response · Authors · 2024-12-01
> > > **To Reviewer uFdg**
> > >
> > > Hi Reviewer uFdg, the discussion period (for reviewers) will end in two days, you have raised some further questions on dataset arrangement and our opinion of paper positioning, which we attach great importance to and agree with your advice.
> > >
> > > (1) We appreciate your advice that we could use the additional evaluations on traditional tabular prediction benchmark (from you recommended paper and conducted in response of "Q1&A1") as the main experiment table with the original evaluations on medical tabular datasets as auxiliary tasks posed in Appendix, we are rearranging the corresponding experiment analysis since the original version is based on the original main evaluations. (2) We agree with your opinion that our method SERSAL would be more appropriate to refer to the setting as unsupervised rather than zero-shot, **to avoid potential confusion and make the paper more friendly to the wider audiences in the future**, we are suitably revising the corresponding main description in the paper, while the paper title can only be revised during camera ready phase. **Overall, the revisions for both (1) and (2) are being carried out simultaneously, and we will reflect your suggested paper expression and organization** in the camera ready version.
> > >
> > > It seems the other reviewers have acknowledged their positive or improved ratings, if necessary, feel free to give your feedback or further questions, thank you very much!

---

> ### Author Response · Authors · 2024-11-24
> **To uFdg (3/4, Q4&Q5&Q6)**
>
> **Q4:** Is TabLLM in Table 2 be tested in a zero-shot scenario? If so, it seems that TabLLM is better than GPT-3.5. Could you please explain the case, given that GPT 3.5 has a more extensive knowledge base compared to TabLLM?
>
> **A4:** Yes, the ‘TabLLM (GPT-3.5)’ baseline in Table 2 is tested in a zero-shot scenario. Actually, there may exist some misunderstanding in experiment settings. As proposed in TabLLM paper [1], TabLLM is a tabular sample serialization and prompt design approach for LLM fine-tuning and prediction, in the end of Sec. 3.2 in TabLLM paper [1], the authors pointed out “TabLLM is both agnostic to the LLM and the specific fine-tuning method that is used. We only consider a single LLM for most of our experiments” and they uniformly used T0 base LLM in their paper. Similarly, in Table 2 we arrange ‘TabLLM (GPT-3.5)’ baseline group using the GPT-3.5 as the base LLM with the same tabular serialization and prompting method in TabLLM paper. Therefore, the group ‘TabLLM (GPT-3.5)’ is better than ‘0-shot (GPT-3.5)’ means TabLLM tabular serialization and prompting method is better than simple 0-shot prompting template, both groups using GPT-3.5 as base LLM.
>
> **Reference:**
>
> [1] TabLLM: Few-shot Classification of Tabular Data with Large Language Models, AISTATS 2023.
>
> ---
>
> **Q5:** What is the meaning of "TabLLM + SERSAL (GPT-3.5)" in Table 2, what is the role of each component?
>
> **A5:** As explained in **Q4&A4**, TabLLM is a general approach to serializing tabular entries (samples) and prompt designing for LLM fine-tuning & prediction on tabular data, which is agnostic to the specific LLM it used. Therefore, group "TabLLM + SERSAL (GPT-3.5)" means we replace simple 0-shot prompting template with TabLLM’s serialization & prompting scheme to query GPT-3.5, bringing a better annotation outcome and improving SERSAL performances. This group indicates our SERSAL can be jointly combined with advanced textual prompting methods (e.g., TabLLM) to achieve further progress since they promise a better LLM annotation quality, SERSAL works in a distinct nature compared to such existing tabular textual prompting methods, forming an orthogonal technical landscape.
>
> According to Q4&Q5, we realize the need to further refine the explanation of the baseline settings and clarify the technical orthogonality of SERSAL in the final paper version. Thank you for your question.
>
> ---
>
> **Q6:** Recent generative models pertained on a large set of tabular data [1] should be considered as baselines since they can be seen as powerful tools for zero-shot performance.
>
> **A6:** Thank you for the insightful suggestion, the LLMs further generatively pre-trained on large tabular data from a wide range of domains are indeed advanced and strong baselines for zero-shot tabular prediction tasks. We additionally evaluate your mentioned paper, the recently released GTL-enhanced LLaMA-2 model (LLaMA-13B-GTL) using its T-lang tabular serialization template [1] on 10 medical diagnosis datasets in Table 2. Notably, the pre-training corpus of the LLaMA-13B-GTL contains 34 tabular datasets from healthcare domain (pointed out in Fig. 5 of its paper [1]), making it powerful in our evaluated medical data. The AUC scores are as follows.
>
> | Dataset:                        |   HF  |   LC  |  ECD  |   LI  |   HE   |  PID  |   FH  |   ST  |   CO  |   AN  |
> |---------------------------------|:-----:|:-----:|:-----:|:-----:|:------:|:-----:|:-----:|:-----:|:-----:|:-----:|
> | supervised FT-Transformer       | 88.19 | 86.61 | 99.60 | 78.94 | 100.00 | 84.72 | 66.25 | 82.98 | 99.91 | 99.92 |
> | 0-shot prompting(GPT-3.5)       | 71.88 | 78.87 | 85.71 | 76.81 |  68.51 | 73.12 | 60.32 | 63.01 | 82.60 | 90.43 |
> | 0-shot prompting(LLaMA-13B-GTL) | 76.26 | 80.73 | 87.06 | 74.63 |  72.83 | 75.61 | 63.72 | 68.47 | 81.96 | 92.38 |
> | SERSAL(0-shot, GPT-3.5)         | 91.39 | 85.42 |  86.40 | 79.39 |  85.14 | 78.97 | 63.97 | 76.36 | 96.85 | 98.37 |
> | SERSAL(0-shot, LLaMA-13B-GTL)   | 93.42 | 87.18 | 88.32 | 78.57 |  90.28 | 80.62 | 66.58 | 79.81 | 95.27 | 99.16 |
>
> As expected, the results of LLaMA-13B-GTL (0-shot prompting) are often better than the ones of GPT-3.5 on evaluated datasets, and as a result, based on the more powerful LLM, the performance of SERSAL is also continuously improved. We will curate all additional results and analysis in our final paper version.
>
> **Essentially, SERSAL is a general loop of thought prompting framework with high flexibility and adaptability, all components including base LLM (e.g., LLaMA-13B-GTL [1]), prompting strategy for annotation (e.g., TabLLM), small model type (e.g., FT-Transformer) can be decided by users according to real-world practice, and the main target of SERSAL is to unconditionally improve tabular prediction results of the selected base LLM, all achieved in an unsupervised manner.**
>
> **Reference:**
>
> [1] From Supervised to Generative: A Novel Paradigm for Tabular Deep Learning with Large Language Models, KDD 2024.

---

> ### Author Response · Authors · 2024-11-24
> **To uFdg (4/4)**
>
> We sincerely appreciate your insightful questions & constructive suggestions that help us improve the paper deeply, and hope our response can address your major concerns and confusions. If you have any further questions or concerns, feel free to raise them and discuss. Thank you very much.

---

> ### Comment · Reviewer_uFdg · 2024-11-26
>
> Many thanks to the authors.
> 1. In the additionally provided results, the performance of 0-shot prompting and SERSAL is 69.05 and 79.36 on AD dataset respectively. However, as illustrated in Table 5 in original paper, the performance is 75.10 and 88.72 respectively. It seems that 69.05 and 79.36 is from Credit dataset?
>
> 2. I am still concern about the position of the paper. Since the authors identify that the position is traditional tabular prediction, the main table, i.e. Table 2 in original paper, can not be used to verify the effectiveness of the proposed method, since they are all medical diagnosis datasets. The comparison between the proposed method and full baseline methods mentioned in the paper on traditional tabular benchmark should be placed in the main table, while the results on medical diagnosis datasets could be used as auxiliary tasks. If the authors complete this, the position of the paper can be more cleared. If not, it is difficult to verify the effectiveness of the proposed method on traditional tabular prediction tasks.

---

> > ### Author Response · Authors · 2024-11-26
> >
> > 1. We appreciate your careful observation of the mistaken performances of Adult (AD) dataset, we did misuse the performances from Credit datasets. We have fixed the mistaken results in the response. Thank you!
> > 2. We agree with your suggestions on using the traditional tabular data benchmark as the main evaluations in the paper, and as we additionally conducted in the "Q1&A1", the SERSAL also works on the non-medical tabular data and multi-class & regression task types. We are rearranging the results of "Q1&A1" and Table 5 as the main evaluation table in the final version, while our current main analysis is based on the medical tabular data, therefore the rearrangement will take some time to carefully  complete, and we will put the current main evaluation on medical datasets to the Appendix as auxiliary tasks.
> >
> > Thank you for your constructive suggestions!

---

### Official Review · Reviewer_jQZe · 2024-11-03

**Soundness:** 4
**Presentation:** 4
**Contribution:** 4
**Rating:** 8
**Confidence:** 4

**Summary:**

The paper proposes a new method, SERSAL, using synergy learning to improve LLMs for zero-shot tabular prediction. SERSAL designs a loop which includes (1) soft LLM pseudo labelling (2) small models training with DivideMix (3) quality control (4) LLM fine-tuning. With one-loop training, SERSAL demonstrates new state-of-the-art zero-shot performance on 9 out of 10 tabular prediction datasets in the medical domain and 3 other binary classification datasets in other domains. Experiments also demonstrate improvements with more (3 loops) training by SERSAL.

**Strengths:**

This paper is overall very well written with clear methodology, experiments, and strong results.

The proposed method SERSAL designs a loop to enhance LLM capabilities in zero-shot tabular prediction. The idea of incorporating the semi-supervised learning DivideMix to address/mitigate noisy labels from LLMs and improve the performance of LLMs with small models (which are trained by DivideMix) is very interesting. Experiments also demonstrate the superior performance of SERSAL and the potential for further improvement with more loops of training, and better foundation LLMs (e.g. GPT-3.5 -> GPT4) across multiple domains.

Superior zero-shot performance across multiple datasets across multiple domains. Reasonable design and setup of experiments.

**Weaknesses:**

The type of tasks is restricted to binary classification. The method may have potential in more complicated tasks and more types of input data (beyond just tabular data) but these are not demonstrated or discussed in this paper.

The experiments on the effectiveness of multi-loop SERSAL are quite light (compared to experiments with one-loop SERSAL). Experiments with more loops and more datasets will be useful.

Elaboration on DivideMix in Section 3 will help improve the readability and understanding of the methodology.

**Questions:**

1. Transformer is still a relatively large model. It will be useful to see how small models like CNN, RNN, or even RF, SVM may work and accelerate the loops.

2. Table 4, please consider changing Loop 0,1,2 to Loop 1,2,3 to avoid confusion.

3. Table 4, should the performance of N-loop SERSAL be the same as (N+1)-loop 0-shot LLM, because (N+1)-loop 0-shot LLM is the fine-tuned LLM from N-loop SERSAL?

---

> ### Author Response · Authors · 2024-11-24
> **To jQZe (1/3, Weakness 1&2)**
>
> We sincerely thank you for your valuable comments and suggestions to uncover more potential of our work, and finding our work robust, significant, interesting with high regard.
>
> ---
>
> **W1:** The type of tasks is restricted to binary classification. The method may have potential in more complicated tasks and more types of input data (beyond just tabular data) but these are not demonstrated or discussed in this paper.
>
> **A1:** We appreciate your sensitive insight of finding untapped potential of our SERSAL, and yes, as you can see, SERSAL is principally not limited to tabular data modality and binary classification task type. Here we add extra evaluation on 3 clinical note datasets (medical text data): MIMIC-III [1] (top-5 label version, i.e., only retain the most frequent 5 ICD code labels) multi-label text classification evaluated with Macro-F1 and two real-world multi-class text classification datasets (Dermatology & Gastroenterology) from [2] evaluated with accuracy score. In these evaluations, we substitute FT-Transformer for tabular data with BERT-base for textual data and use single-loop SERSAL, and report the detail results as follows:
>
> | clinical note   datasets: | MIMIC-III(top5) | Dermatology  | Gastroenterology |
> |---------------------------|:---------------:|:------------:|:----------------:|
> | task type                 |   multi-label   |  multi-class |    multi-class   |
> | Supervised BERT-base      |      60.45      |     77.89    |       69.75      |
> | 0-shot prompting(GPT-3.5)           |      62.08      |     63.75    |       46.65      |
> | CoT(0-shot, GPT-3.5)              |      68.25      |     70.36    |       55.73      |
> | SERSAL(0-shot, GPT-3.5)           |      65.33      |     72.18    |       61.89      |
> | SERSAL(0-shot, GPT-4)             |      72.43      |     76.92    |       68.52      |
>
> As we can see, SERSAL is also applicable to the textual data and more than binary classification. We also provide the case of tabular regression tasks in the reply Q1&A1 of Reviewer uFdg. Since the traditional textual (linguistic) prompting methods (e.g., CoT) is effective for textual data but not usable for tabular data, that is why our paper primarily focus on tabular data perspective.
>
> **Reference:**
>
> [1] MIMIC-III, a freely accessible critical care database, Scientific data 2016.
>
> [2] Text2Tree: Aligning text representation to the label tree hierarchy for imbalanced medical classification, EMNLP Findings 2023.
>
> ---
>
> **W2:** The experiments on the effectiveness of multi-loop SERSAL are quite light (compared to experiments with one-loop SERSAL). Experiments with more loops and more datasets will be useful.
>
> **A2:** Thank you for the question, we agree that conducting more multi-loop SERSAL evaluations on other datasets can help to demonstrate the usability of SERSAL better. We originally evaluated on ECD and LI datasets with 3 loops in Table 4 since: (1) The performances of these two datasets are close to the ones of baseline, thus we choose them for multi-loop evaluation to inspect the further potential of SERSAL; (2) To save the cost of ChatGPT API. Here we add extra evaluations of multi-loop SERSAL on 4 datasets (6 datasets in total) with data scale smaller than 1000 samples as follows, using 5-loop SERSAL and AUC score as metric:
>
> | # Loop                                       |   HF   |   LC   |   ECD  |   LI   |   HE   |   PID  |
> |----------------------------------------------|:------:|:------:|:------:|:------:|:------:|:------:|
> | N/A (original 0-shot prompting GPT-3.5) | 71.88  | 78.87  | 85.71  | 76.81  | 68.51  | 73.12  |
> | 1-loop                                            | 91.39  | 85.42  | 86.40  | 79.39  | 85.14  | 78.97  |
> | 2-loop                                            | 92.03  | 86.15  | 87.00  | 82.47  | 87.32  | 80.61  |
> | 3-loop                                            | 93.58  | 85.42  | 89.00  | 84.07  | 89.57  | 81.83  |
> | 4-loop                                            | 93.58  | 86.15  | 89.60  | 84.98  | 89.93  | 82.37  |
> | 5-loop                                            | 93.58  | 86.15  | 89.20  | 84.98  | 89.93  | 82.72  |
>
> As we can see, the most performance gain is acquired at the first-loop SERSAL process, showing the significant bonus of learning from the LLM annotations to refine the inherent knowledge at the first time, and further improvement is minor and tends to gradually diminish after around 3 loops. The results suggest that in practical usage, if the computational budget is limited, a single-loop SERSAL enough to obtain considerable benefits. We will curate the results and analysis in the final paper version.

---

> ### Author Response · Authors · 2024-11-24
> **To jQZe (2/3, Weakness 3&Q1)**
>
> **W3:** Elaboration on DivideMix in Section 3 will help improve the readability and understanding of the methodology.
>
> **A3:** Thank you for your nice suggestion on making our paper more friendly to the audiences from different background. The Reviewer diWx also gives the similar suggestion in W1&Q1, and we have appended the detailed elaboration of how DivideMix works and its usage in SERSAL to the Appendix D of the paper. Here we give the explanation of its mechanism:
> In the traditional noisy data learning field, it was theoretically proved and empirically observed that the “memorization” behavior of neural networks leads to different optimization behavior on real data and noisy ones that neural networks tend to learn simple patterns first before fitting label noise [1]. Based on this theoretical foundation, a typical group of noisy label learning methods [2,3,4] exploit per-sample training loss to judge the noisy labels, for example, in our paper we adopt DivideMix [4] to learn the small model from LLM noisy annotations, which models the noise probabilities of each sample by dynamically fitting a Gaussian Mixture Model (GMM) on per-sample losses, all training samples are divided into a clean set and a noisy set based on a probability threshold. During the DivideMix training process, samples in the clean set are used for supervised learning (using their LLM annotations), while ones in the noisy set is used in an unsupervised manner (only using their features), e.g., regularization loss or reconstruction task. The process will be ended until the average loss of heuristically selected early stopping subset (high-LLM-confidence samples) is converged. Notably, clean sample is not equal to high-LLM-confidence samples, but the sample which is easier to fit by the model. Since the small model (i.e., FT-Transformer here) is only supervised by clean data and regularized on noisy data, all data is fully and reasonably exploited to acquire a better pattern.
>
> **Reference:**
>
> [1] A closer look at memorization in deep networks, ICML 2017.
>
> [2] Mixmatch: A holistic approach to semi-supervised learning, NeurIPS 2019.
>
> [3] Unsupervised label noise modeling and loss correction, ICML 2019.
>
> [4] DivideMix: Learning with noisy labels as semi-supervised learning, ICLR 2019.
>
> ---
>
> **Q1:** Transformer is still a relatively large model. It will be useful to see how small models like CNN, RNN, or even RF, SVM may work and accelerate the loops.
>
> **A4:** Here we additionally inspect the impact of the small model selection of SERSAL by replacing the FT-Transformer with ResNet (CNN-based tabular model) using configs in [1].
>
> | Dataset:                       |   HF  |   LC  |  ECD  |   LI  |   HE  |  PID  |   FH  |   ST  |   CO  |   AN  |
> |--------------------------------|:-----:|:-----:|:-----:|:-----:|:-----:|:-----:|:-----:|:-----:|:-----:|:-----:|
> | 0-shot prompting(GPT-3.5)      | 71.88 | 78.87 | 85.71 | 76.81 | 68.51 | 73.12 | 60.32 | 63.01 | 82.60 | 90.43 |
> | SERSAL(GPT-3.5+FT-Transformer) | 91.39 | 85.42 |  86.4 | 79.39 | 85.14 | 78.97 | 63.97 | 76.36 | 96.85 | 98.37 |
> | SERSAL(GPT-3.5+ResNet)         | 90.32 | 84.86 | 86.23 | 78.88 | 86.78 | 76.39 | 63.51 | 75.83 | 96.26 | 97.68 |
>
> It can be seen that even though SERSAL using FT-Transformer is slightly better, the overall significance of applying SERSAL at the first time still holds which is not impacted by the used small model. We also list the consuming time of each step in in a single SERSAL loop in the HF (Heart Failure Prediction) dataset (303 samples) as follows:
>
> | SERSAL step in HF dataset | GPT-3.5 annotation(per sample) | GPT-3.5 annotation | DivideMix(FT-Transformer) | DivideMix(ResNet) | GPT-3.5 fine-tuning |
> |---------------------------|:------------------------------:|:------------------:|:-------------------------:|:-----------------:|:-------------------:|
> | Time Usage (second)       |               2.3              |         697        |             28            |         16        |         126         |
>
> As we can see, in a single loop, LLM annotation and fine-tuning, especially using the online blackbox LLMs like GPT-3.5&4 used in this paper, occupy the major part of the SERSAL training time, which is dominantly impacted by the real-time network latency and underlying optimization engineering of ChatGPT APIs. While once the SERSAL process is finished, its inference on the small model is very fast. We will curate the results and add the analysis in the final version.
>
> **Reference:**
>
> [1] Revisiting Deep Learning Models for Tabular Data. NeurIPS 2021.

---

> ### Author Response · Authors · 2024-11-24
> **To jQZe (3/3, Q2&3)**
>
> **Q2:** Table 4, please consider changing Loop 0,1,2 to Loop 1,2,3 to avoid confusion.
>
> **A5:** Thank you for the advice! We have changed the corresponding denotation to avoid the confusion.
>
> ---
>
> **Q3:** In Table 4, should the performance of $N$-loop SERSAL be the same as ($N+1$)-loop 0-shot LLM, because ($N+1$)-loop 0-shot LLM is the fine-tuned LLM from $N$-loop SERSAL?
>
> **A6:** The LLM of $N$-loop SERSAL denotes the one before fine-tuning in this loop, for example, after changing the denotation in Q2&A5, 1-loop LLM means the original GPT-3.5, 2-loop LLM means the GPT-3.5 fine-tuned after the first loop. $N$-loop SERSAL results are calculated with the small model outputs, for example, 1-loop SERSAL means the performance of the small model in the first loop. Therefore, in practice, if users only perform single-loop SERSAL, it is no need to fine-tuning the LLM, just using the small model for inference, which can further save the training time.
>
> ---
>
> We hope our response can address your questions and confusions, if you have further questions, feel free to raise them and discuss.

---

> > ### Comment · Reviewer_jQZe · 2024-11-26
> >
> > Thanks to the authors for the extensive responses!

---

> > > ### Author Response · Authors · 2024-11-28
> > > **Thank you for the positive evaluation and value recognition**
> > >
> > > We profoundly thank you for the positive evaluations on the novelty, quality and significance of our paper. SERSAL is another fundamentally novel technical landscape using learnable small models to probe the untapped capability of LLMs, that leaves a large space to explore and is orthogonal to prevailing textual prompting techniques. The strong results and continuous improvement (as more powerful LLMs appear) indicate its great potential and benefit for the current LLM community, especially on the tabular data prediction tasks.

---

### Official Review · Reviewer_diWx · 2024-11-03

**Soundness:** 4
**Presentation:** 3
**Contribution:** 4
**Rating:** 8
**Confidence:** 4

**Summary:**

This study focused on training a small model to effectively promote LLMs to enhance their zero-shot predictions performance on healthcare tabular data. The study was motivated by the observations that, in the literature, most of the related studies have been primarily focused on either the medical imaging data or the text data. While, healthcare tabular data may pose their unique challenges, where it is common to have widely heterogenous distributions across different variables. The proposed idea of synergy learning is to first deploy LLM to generate some noisy annotations to allow the small model to learn and generate tuning promotes to refine LLM for the task; and we iterate this "synergy learning" to continuous improve the model's final performance. The authors conducted detailed experiments with some publicly healthcare tabular datasets to validate the performance of their proposed approach. Additionally, the authors have also investigated the generalizability of the proposed approach over other domains via evaluations with 3 classification datasets and 1 generated dataset, aiming to simulate the scenario for a unknown domain.

**Strengths:**

1/ The manuscript is, in general, well written, Objectives and motivations of the study were clearly explained. Details of the experiments were provided. Key findings were appropriately summarized in tables and figures.

2/ I share the same belief that the size of the LLMs continues to go. It will become infeasible to retrain or fine tune the entire LLMs for specific domains. The use of small models to learn from LLMs and/or to promote LLMs is indeed a promising direction to explore.

3/ The study proposed an interesting synergy learning mechanism. In the proposed mechanism, small model first learn from the noisy annotations from LLMs with low confidence, then, after quality controls, the learned small model generate promotes aiming to refine LLMs' predictions; while the refined LLMs will again generate slightly more confident labels to further train the small model. The proposed synergy learning process is interesting and sound.

4/ The team has evaluated the performance of the proposed approach over 10 healthcare datasets focusing on different disease. This helps to show the stability of the proposed approach's performance over healthcare tabular data.

5/ Also, to demonstrate the potential generalizability of the proposed approach to other domains (other than healthcare), it was also evaluated over 3 binary classification datasets and also 1 simulated dataset.

Overall the quality of the manuscript is good. Nevertheless, there are still some areas for improvements as mentioned next.

**Weaknesses:**

1/ Better explanation on background knowledge for our general LLM audiences

The proposed SERSAL prompting for LLMs on tabular prediction task bears an essentially different mechanism compared to prevailing textual prompting methods, requiring relevant knowledge on noisy label learning theory for general audiences in LLM research fields, it will be better to detail the inherent mechanism on why a better small model can be obtained by learning from the LLM noisy outputs.

2/ Dependency on initial LLM performance?

As pointed out in the Limitation Section (Appendix A), the proposed SERSAL prompting requires the LLMs with latent knowledge in the target domain to be effective, though a randomly generated dataset (Fake dataset in Table 5) is evaluated to partially answer SERSAL unusability on unreasonable data domain, but the case of reasonable data with very low initial accuracy and how to handle such datasets is not discussed (the fault tolerance limit or the least initial LLM performance that SERSAL can be accepted).

3/ Lack of detailed computational complexity analysis

The current manuscript did not analyze the computational complexity in detail which may be important for the usability of SERSAL on large-scale datasets. As one of the key novelty of the study is to enable the use of small models to enhance the learning efficiency of large models. Do think that detailed computational complexity studies are needed to justify this point.

4/ Some reference formatting issue

There are several instances of the misuse of LaTeX citation commands `\citep` and `\cite` in the paper (e.g., Line 65, 76, 85 and the caption of Fig. 1).

Please carefully correct the relevant LaTeX commands during the rebuttal phase.

**Questions:**

1/ Point 1 under weaknesses

It is interesting to aggregate LLM knowledge in a small tabular model in SERSAL process, could you give a detailed mechanism explanation on why a better performed small model can be learned from LLM outputs?

2/  Point 2 under weaknesses

Wonder how sensitive the effectiveness of SERSAL prompting is to the initial LLM performance on the target dataset, or how to apply SERSAL on the data domain where the initial performance is not good?

3/ Point 3 under weaknesses

Could provide the detailed computational complexity analysis of SERSAL in its real-world applications?

---

> ### Author Response · Authors · 2024-11-24
> **To diWx (1/2, Q1&Q2)**
>
> We sincerely thank you for your significant advice, careful reviewing and appreciate for finding our work high-quality, promising, sound and interesting.
>
> ---
>
> **W1&Q1:** Require wide knowledge for the general audiences, could you give a detailed mechanism explanation on why a better performed small model can be learned from LLM outputs?
>
> **A1:** Thank you for your suggestion, here we detail the mechanism explanation of how to learn a better small (neural network) model from LLM noisy annotations. In the traditional noisy data learning field, it was theoretically proved and empirically observed that the “memorization” behavior of neural networks leads to different optimization behavior on real data and noisy ones that neural networks tend to learn simple patterns first before fitting label noise [1]. Based on this theoretical foundation, a typical group of noisy label learning methods [2,3,4] exploit per-sample training loss to judge the noisy labels, for example, in our paper we adopt DivideMix [4] to learn the small model from LLM noisy annotations, which models the noise probabilities of each sample by dynamically fitting a Gaussian Mixture Model (GMM) on per-sample losses, all training samples are divided into a clean set and a noisy set based on a probability threshold. During the DivideMix training process, samples in the clean set are used for supervised learning (using their LLM annotations), while ones in the noisy set is used in an unsupervised manner (only using their features), e.g., regularization loss or reconstruction task. The process will be ended until the average loss of heuristically selected early stopping subset (high-LLM-confidence samples) is converged. Notably, clean sample is not equal to high-LLM-confidence samples, but the sample which is easier to fit by the model. Since the small model is only supervised by clean data and regularized on noisy data, all data is fully and reasonably exploited to acquire a better pattern.
>
> We have appended the explanation details in the Appendix D of the paper to help audiences in different backgrounds friendly understand inherent mechanism of SERSAL.
>
> **Reference:**
>
> [1] A closer look at memorization in deep networks, ICML 2017.
>
> [2] Mixmatch: A holistic approach to semi-supervised learning, NeurIPS 2019.
>
> [3] Unsupervised label noise modeling and loss correction, ICML 2019.
>
> [4] DivideMix: Learning with noisy labels as semi-supervised learning, ICLR 2019.
>
> ---
>
> **W2&Q2**: Dependency on initial LLM performance? How sensitive SERSAL is to the initial LLM annotation performance on the target dataset, or how to apply SERSAL on the data domain where the initial performance is not good?
>
> **A2:** Thank you for the insightful question. We agree that the randomly generated “Fake” dataset in Table 5 only considers the case where the LLM has no knowledge in the target domain, which is not equivalent to the case that the LLM has poor domain knowledge, therefore, we arrange evaluation on difficult datasets from clinical trial domain (mouse mortality prediction based on the new drug trial data) in Table 7 and discuss in Sec. 3.2 “Continuous Performance Growth”. As we can see, though initial 0-shot GPT-3.5 performs poorly on dataset “N00041119” (56.79% AUC) & “N00079274” (59.85% AUC), using SERSAL (group “SERSAL (GPT-3.5)”) can still acquire performance boost (become 58.31% and 64.17% after SERSAL), and the overall enhancement effect is positively correlated with the initial performance.
>
> **How to apply SERSAL on the data domain where the initial performance is not good?** For example, if the initial AUC is lower than 56.79% or even 50%, according to the group “SERSAL (GPT-4)” in Table 7, we think a fundamental solution is to adopt more powerful base LLMs, as the LLM community develops, our SERSAL will be more powerful continuously, in practice, the users can choose or fine-tune domain-adapted LLMs or more strong base LLMs to avoid poor initial performance issues, and our SERSAL is positioned at unconditionally improve their prediction results on tabular data.
>
> **What is the fault tolerance limit or the least initial LLM performance that SERSAL can be accepted?** Actually this question can be reformulated as “what is the maximum noise rate the DivideMix (or other noisy label learning methods) can tolerate?”, which is still an unexplored theory problem yet in the noisy data learning field and requires further research and exploration separately in the future.

---

> ### Author Response · Authors · 2024-11-24
> **To diWx (2/2, Q3)**
>
> **W3&Q3:** Could you provide the detailed computational complexity analysis of SERSAL in its real-world applications?
>
> **A3:** Of course, we analyze the computational time of SERSAL in detail as follows: During training, a complete single SERSAL loop includes three computational costs, i.e., (a) LLM annotation for tabular data, (b) noisy data learning with the small model, (c) fine-tuning the LLM with small model outputs. For step (a), we stack several samples into one query to further reduce the number of requests to the online blackbox ChatGPT, assume the data scale is $N$, each query contains $k$ samples and costs time $t$, then the total time of part (a) is $T_a = \frac{N \times t}{k}$, $t$ is majorly affected by real-time network latency and underlying optimization engineering of ChatGPT APIs rather than context size. For step (b), in this paper we train a FT-Transformer (in default config [1]) with DivideMix [2], the time of part (b) is equivalent to fine-tuning an FT-Transformer (3-layer Transformer) which is significantly smaller than the time of part (a). For step (c), the time is from fine-tuning the ChatGPT with given API using $N$ textualized tabular data annotated by the FT-Transformer. If we just perform single-loop SERSAL, only step (a) and (b) are required. During inference, we directly use the small model which inference cost is very small compared to training phase.
>
> **Reference:**
>
> [1] Revisiting Deep Learning Models for Tabular Data. NeurIPS 2021.
>
> [2] DivideMix: Learning with noisy labels as semi-supervised learning, ICLR 2019.
>
> ---
>
> **W4:** Some reference formatting issue? There are several instances of the misuse of LaTeX citation commands.
>
> **A4:** Thank you for carefully checking the reference formatting in the paper, we have fixed all the misuse LaTex commands and check there is no formatting issue now.
>
> ---
>
> We hope our response can address your questions. If you have further questions, feel free to raise them and discuss.

---

> > ### Comment · Reviewer_diWx · 2024-12-01
> >
> > Thanks for the details response to my questions.

---

> > > ### Author Response · Authors · 2024-12-03
> > > **Thank you for the positive recognition**
> > >
> > > Thank you for your feedback and positive evaluations. We are glad we have addressed your questions.

---

### Official Review · Reviewer_m9i1 · 2024-11-04

**Soundness:** 2
**Presentation:** 2
**Contribution:** 2
**Rating:** 5
**Confidence:** 4

**Summary:**

The authors propose a framework for refining a pretrained LLM for zero-shot tabular prediction. In particular, the authors propose to iteratively refine a given LLM's tabular prediction capability via a small "student" tabular prediction model (FT-Transformer), where at each iteration, the LLM is prompted to generate noisy labels for an unlabeled training dataset, the tabular prediction model is trained on the noisy labels based on the DivideMix approach, and the predictions from the trained tabular prediction model is used to fine-tune the LLM. Focusing primarily on tabular prediction datasets from the medical domain, the authors conduct several experiments using GPT-3.5 and GPT-4, and compare the resulting performance of their approach against those of alternative prompting methods.

**Strengths:**

- The limited generalizability/transferability of LLMs to the tabular domain, which is discussed as the main motivation for the paper, is a well-known issue and an active area of research.
- The idea of using an LLM as noisy labelers and combining it with semi-supervised learning methods designed to handle noisy labels is  interesting.
- Several ablation experiments are considered to investigate the impact of different design choices (e.g., soft vs. hard labels from the LLM, early stopping), and empirically validate the assumption that high LLM confidence scores tend to correlate with the accuracy of the LLM-generated labels.

**Weaknesses:**

- Overall, there appears to be a misalignment in the motivation for the paper (i.e., improving the zero-shot generalizability of LLMs for tabular prediction) and what is proposed in the paper. My interpretation of what is really being addressed in the paper is "How can we leverage LLMs as noisy labelers to train a downstream *tabular prediction model* (e.g., FT-Transformer) when we only have access to an *unlabeled* training set. In particular, the following are my specific concerns:
    - While the proposed iterative process also involves fine-tuning the LLM, predictions on any test set are generated using the downstream tabular prediction model, as presented in line 13 of Algorithm 1 ($\theta^{*(t)}$ denotes the parameter of the downstream model and not the LLM).
    - The proposed approach is not really "zero-shot" in that while the training dataset does not include the ground-truth labels, the dataset is still being used to explicitly train the tabular prediction (and also the LLM for that matter).
    - The Introduction discusses limitations of LLMs in handling heterogeneous numerical features, but the paper does not directly address how to resolve this problem for LLMs.
    - As such, the title and the overall framing of the work appear ill-conceived, as the work is not really enhancing the "zero-shot tabular prediction for LLMs".
- In Section 2.2, the overall description of how the DivideMix approach is being applied to the considered context would benefit from better clarity and details (even if pushed to the Appendix given space constraints). For example, how is the early stopping being performed within the adapted DivideMix algorithm? What are the hyperparameters in the phrase "hyper-parameter selection for the teaching process" and how are they being selected in the absence of a validation set?
- In the experiments, the proposed approach is compared against alternative zero-shot (vanilla zero-shot, zero-shot CoT, TabLLM, LIFT) and few-shot prompting strategies for generating tabular predictions from off-the-shelf LLMs, which do not seem to be the most relevant baselines here. The proposed approach involves directly *fine-tuning* the LLM as opposed to being a new "prompting" strategy (not to mention that it's not even really the LLM that's generating the final predictions), and while TabLLM and LIFT were both originally proposed for fine-tuning the LLM for tabular prediction, these baselines are only considered in the zero-shot prompting regime.
    - On a related note, since TabLLM and LIFT were considered in the zero-shot setting, this would mean that the only difference between them and the vanilla zero-shot prompting baseline is really in the prompt format and the set of instructions. The AUC values in Table 2 seem to indicate that the prediction results are highly sensitive to such choices. It is well-known in the literature that LLMs are highly sensitive to the choices of the prompt [1] and the tabular feature serialization method [2,3], so the results should appropriately account for these concerns to ensure a fair comparison.
- The datasets considered in the main evaluations (Section 3.1) are publicly available datasets, with most of them having been around for a long time. As such, there is a potential risk that these datasets have been part of the LLM pretraining corpus, which limits the generalizability of the findings. While the authors do conduct an ablation study in Section 3.6 involving a randomly generated dataset (denoted as "Fake") and demonstrate that the proposed approach can fail if the model does not have relevant domain knowledge, it seems important that the evaluations are also carried out on either (i) private datasets or (ii) public datasets that have been released after the pretraining-data cutoff dates for the LLMs, which still come from a domain that the LLMs would carry relevant domain knowledge. For example, would the LLM still generate high-quality confidence scores if we were to test on a new medical diagnosis dataset released in the future? It appears important that the LLM confidence scores are reasonably well-calibrated for this approach to work in the first place; otherwise, the error will only compound over several iterations of the proposed algorithm.
- In Figure 2 and Table 4, why are the results shown only for the ECD and LI datasets? Do similar trends hold for the other datasets?

References:

[1] Quantifying Language Models' Sensitivity to Spurious Features in Prompt Design or: How I learned to start worrying about prompt formatting (Sclar et al., 2024)

[2] TabLLM: Few-shot Classification of Tabular Data with Large Language Models (Hegselmann et al., 2023)

[3] Large Language Models(LLMs) on Tabular Data: Prediction, Generation, and Understanding -- A Survey (Fang et al., 2024)

**Questions:**

Please see my comments in the Weaknesses section.

---

> ### Author Response · Authors · 2024-11-26
> **To m9i1 (1/6, Weakness 1)**
>
> **W1:** Motivation misalignment?
>
> **A1:** Thank you for your different motivation understanding and careful consideration, we would like to reply your concern from your mentioned **four points**.
>
> (1) SERSAL prediction is based on the small downstream tabular prediction model, as presented in the line 13 of Algorithm 1 ($\theta^{*(t)}$ denotes the parameter of the small tabular prediction model fitted from LLM annotations), since the prediction is not directly based on the LLM, it should not be interpreted as “improving the zero-shot generalizability of LLMs for tabular prediction”?
>
> **A1-1:** As you pointed out, we use the small tabular model for SERSAL prediction, and the small model is trained from noisy LLM annotations, therefore, the prediction ability and performance of the small model is also an extension of, or, from the LLM capability itself, without access to any external knowledge. Technically speaking, there exists similar inspiration that introduces external learnable parameters for better LLM prediction, such as LoRA, while SERSAL is adapted to the blackbox LLM (LLM parameters and logits inaccessible) and learning from the LLM itself. Here we list the detail prediction performance of the GPT-3.5 fined-tuned by the small model outputs after each loop/iteration.
>
> | # Loop                                              |   HF   |   LC   |   ECD  |   LI   |   HE   |   PID  |
> |-----------------------------------------------------|:------:|:------:|:------:|:------:|:------:|:------:|
> | 0-shot prompting   GPT-3.5(original)                | 71.88  | 78.87  | 85.71  | 76.81  | 68.51  | 73.12  |
> | SERSAL(predicted by the   FT-Transformer, 1st loop) | 91.39  | 85.42  | 86.40  | 79.39  | 85.14  | 78.97  |
> | 0-shot prompting  GPT-3.5(fine-tuned by 1st FT-T)   | 87.58  | 83.64  | 86.42  | 80.26  | 86.18  | 79.26  |
> | SERSAL(predicted by the   FT-Transformer, 2nd loop) | 92.03  | 86.15  | 87.00  | 82.47  | 87.32  | 80.61  |
> | 0-shot prompting  GPT-3.5(fine-tuned by 2nd FT-T)   | 89.26  | 85.39  | 87.81  | 82.91  | 86.87  | 81.47  |
> | SERSAL(predicted by the   FT-Transformer, 3rd loop) | 93.58  | 85.42  | 89.00  | 84.07  | 89.57  | 81.83  |
> | 0-shot prompting  GPT-3.5(fine-tuned by 3rd FT-T)   | 91.03  | 84.67  | 89.24  | 84.32  | 88.46  | 81.72  |
> | SERSAL(predicted by the   FT-Transformer, 4th loop) | 93.58  | 86.15  | 89.60  | 84.98  | 89.93  | 82.37  |
> | 0-shot prompting  GPT-3.5(fine-tuned by 4th FT-T)   | 91.03  | 84.67  | 89.24  | 84.63  | 88.46  | 82.08  |
> | SERSAL(predicted by the   FT-Transformer, 5th loop) | 93.58  | 86.15  | 89.20  | 84.98  | 89.93  | 82.72  |
>
> As we can see, after fine-tuned by the small model outputs, the performance of LLM itself is continuously improving, though the small model performs slightly better than the LLM in each loop/iteration. For better understanding, you can treat the small model as a prediction interface of the LLM, which summaries the signal from the LLM and is only a different prediction manner compared to common direct LLM inference. Therefore, we use the small model for prediction for a better result. Also, prediction with the small model can save computational cost during inference and is preferred in online service requirement for real-world tabular prediction applications. We will curate the above results and analysis here to the final paper version to provide reference for future audiences.
>
> (2) SERSAL is not really “zero-shot” even though the training dataset does not include the ground truth labels, since the features of training data are used?
>
> **A1-2:** We appreciate your rigorous consideration, this question is the same as Q2 of Reviewer uFdg. The concept of your mentioned “zero-shot” in this paper is equivalent to “no (ground truth) label”, that is why we use “zero-shot tabular prediction” rather than “zero-shot learning”, and SRESAL is technically in an unsupervised data setting for LLM tabular prediction, which is also pointed out by Reviewer uFdg. The expression “zero-shot tabular prediction” is inspired from “zero-shot prompting” since these textual prompting methods are used as baselines. **We will emphasized the unsupervised nature of SERSAL in title, keywords, abstracts and the method statement to avoid the future confusion**, thank you! (we cannot modify title, keywords at present, they can only be changed during camera ready phase).

---

> ### Author Response · Authors · 2024-11-26
> **To m9i1 (2/6, Weakness 1&2)**
>
> (3) The Introduction section discusses limitations of LLM handling heterogenous numerical tabular features, but the paper does not directly address the problem from this perspective?
>
> **A1-3:** As you point out in the first strength, the inferiority of LLM handling heterogenous numerical tabular data is a well-known issue and our discussion is to introduce this widely recognized background, which does not mean that we are limited to address the LLM inferiority from the perspective of heterogeneous and numerical nature of tabular features. Many great works solve seemingly unlovable problems from a completely new perspective, which also introduce novel technical routes and thinking for the corresponding research fields.
>
> (4) The title and framework of the paper appear ill-conceived, as not really enhancing “zero-shot tabular prediction for LLMs”?
>
> **A1-4: Thank you for your unreserved thinking, judgement on whether the paper title and framework are well express the motivation is subjective that varies in different audiences from different research backgrounds. The comments from other reviewers express their positive value recognition of our work, we respect different voices and that is why we actively discuss and reply here to figure out the disagreement and exchange academic idea. Based on your concern, we have further specified the ability of “LLM zero-shot tabular prediction” here as the performance of LLM on tabular prediction without access to ground truth labels to avoid potential target confusion for the future audiences.**
>
> ---
>
> **W2:** The overall description of how the DivideMix approach is being applied to the considered context would benefit from better clarity and details?
>
> **A2:** We appreciate your constructive suggestion! Similar suggestion is mentioned in the Q1&A1 for Reviewer diWx and weakness 3&A3 for Reviewer jQZe. Based on the suggestions from you and the other two reviewers, **we have detailed the mechanism and usage of DivideMix in our SERSAL process in Appendix D to make the paper more friendly to the audiences**. Here we give a detailed description on its mechanism and hyperparameter settings.
>
> **DivideMix mechanism in SERSAL:** In the traditional noisy data learning field, it was theoretically proved and empirically observed that the “memorization” behavior of neural networks leads to different optimization behavior on real data and noisy ones that neural networks tend to learn simple patterns first before fitting label noise [1]. Based on this theoretical foundation, a typical group of noisy label learning methods [2,3,4] exploit per-sample training loss to judge the noisy labels, for example, in our paper we adopt DivideMix [4] to learn a small model using LLM noisy annotations, which models the noise probabilities of each sample by dynamically fitting a Gaussian Mixture Model (GMM) on per-sample losses, all training samples are divided into a clean set and a noisy set based on a probability threshold $\tau$. During the DivideMix training process, samples in the clean set are used for supervised learning (using their soft LLM annotations), while ones in the noisy set is used in an unsupervised manner (only using their features), e.g., learn with regularization loss or reconstruction task. The process will be ended until the average loss of heuristically selected early stopping subset (high-LLM-confidence samples $D_{es}$ in Algorithm 1) is converged, i.e., the loss of early stopping subset is not decreased for $m$ epochs. Notably, clean sample is not equivalent to high-LLM-confidence sample, but the sample which LLM annotation is easier to fit by the small tabular model. Since the small model (i.e., FT-Transformer here) is only supervised by clean data and regularized on noisy data, all data is sufficiently and reasonably exploited to acquire a better pattern.
>
> **DivideMix hyperparameters in SERSAL:** We refer to the original hyperparameter settings in DivideMix paper [4] and only search the temperature ($T$) in {0.5, 5.0, 10.0}, with fixed regularization loss weight $L_u$ to 25, clean probability $\tau$ to 0.9, and the learning rate of the small model (FT-Transformer) to 1e-4. Additionally, we uniformly introduce the early stopping patience $m$ to 5. The best temperature is selected based on the training loss of early stopping subset $D_{es}$.
>
> **Reference:**
>
> [1] A closer look at memorization in deep networks, ICML 2017.
>
> [2] Mixmatch: A holistic approach to semi-supervised learning, NeurIPS 2019.
>
> [3] Unsupervised label noise modeling and loss correction, ICML 2019.
>
> [4] DivideMix: Learning with noisy labels as semi-supervised learning, ICLR 2019.

---

> ### Author Response · Authors · 2024-11-26
> **To m9i1 (3/6, Weakness 3)**
>
> **W3:** In the experiment, the zero-shot or few-shot prompting baselines are based on the off-the-shelf LLM (i.e., GPT-3.5 here) without fine-tuning, while SERSAL fine-tunes the LLM (though not use any ground truth labels) and the GPT-3.5 used in SERSAL is not the original one, it seems these baselines are not relevant enough (for example, TabLLM and LIFT are originally proposed for supervisedly fine-tuning LLMs on tabular data)? Besides, since TabLLM and LIFT baselines only differ in prompt format, and in Table 2 it seems such different prompting schemes affect the LLM direct prediction on.
>
> **A3:** We very appreciate your careful consideration on baseline settings, to make your advice clearer and more actionable, we break down the question into 3 sub-questions based on our understanding.
>
> (1) Why use the off-the-shelf LLM for evaluation?
>
> **A3-1:** Adapting the capability of LLMs to various real-world applications & data modality has been a research trend as the rapid development of powerful LLMs, especially the commercial online blackbox ones such as GPT-3.5&4. Our SERSAL is usable for such online blackbox LLMs thus we use ChatGPT for experiment to show its flexible adaptability on LLMs (any LLMs with fine-tuning APIs given are compatible).
>
> (2) Although SERSAL does not use ground truth labels, it still fine-tunes the LLM, while the baselines all use non-fine-tuned LLM, it seems the baseline arrangement is not relevant enough?
>
> **A3-2:** As replied in **A1-2** of weakness 1, the mentioned “zero-shot prediction” is equivalent to “no (ground truth) label”, from which perspective we arrange these baselines. **On the one hand**, as a prevailing technical roadmap for directly and unconditionally improving LLM reasoning ability, we want to point out textual prompting (prompt engineering) techniques in tabular prediction tasks are not as successful as their wide applications in unstructured data tasks, and even the recently proposed TabLLM [1] & LIFT [2] are originally proposed for supervisedly fine-tuning LLMs on tabular data, **there exists a counterpart void that can directly and unconditionally improve performance of LLM on tabular prediction without access to ground truth labels**, and that is why we propose SERSAL. **On the other hand**, as a non-textual “prompting” (you can treat SERSAL as a self-prompting, or more precisely, unsupervised learning technique), our SERSAL provides a **fundamentally distinct technical nature**, and there has been no baseline counterpart with similar mechanism in the LLM community, which reflects **orthogonal technique novelty** of our work and is why we cannot find a very relevant baseline.

---

> ### Author Response · Authors · 2024-11-26
> **To m9i1 (4/6, Weakness 3&4)**
>
> (3) From the results of zero-shot TabLLM & LIFT, it seems different prompt formats (prompt engineering schemes) still affect the LLM direct performance on tabular prediction, how to ensure a fair comparison considering the impact of prompt formats?
>
> **A3-3:** Since the space of possible prompt formats (prompt engineering schemes) is infinite, it is impossible to explore all possible schemes, therefore, we arrange two recent representative tabular prompting works, i.e., TabLLM and LIFT in Table 2, and TabLLM was also evaluated in the zero-shot data regime in its paper [1]. As you pointed out, different prompt tricks may lead to different baseline results, which bonus can also be jointly exploited by our SERSAL. Essentially, SERSAL is a general loop of thought prompting framework with high flexibility and adaptability, all components of base LLM (e.g., GPT-3.5&4), prompt engineering scheme for annotation (e.g., TabLLM), small model type (e.g., FT-Transformer) can be decided by users according to real-world practice, and the main target of SERSAL is to unconditionally improve tabular prediction results of the selected base LLM, all achieved in an unsupervised manner. **Therefore, our SERSAL is able to continue to be stronger as more powerful base LLMs and effective prompt engineering scheme appear. The existing tabular prompting baselines do not technically conflict with SERSAL but make it better.** Apart from the results of SERSAL using simple 0-shot prompt templates in the paper, here we additionally conduct SERSAL using TabLLM and LIFT as the prompt schemes to query LLM annotations to provide a better demonstration of the collaborative beneficial relationship between SERSAL and the used tabular prompting baselines.
>
> |                                   |   HF   |    LC    |    ECD   |    LI    |    HE    |    PID   |    FH    |    ST    |    CO    |    AN    |
> |-----------------------------------|:------:|:--------:|:--------:|:--------:|:--------:|:--------:|:--------:|:--------:|:--------:|:--------:|
> | 0-shot prompting(GPT-3.5)         | 71.88  | 78.87    | 85.71    | 76.81    | 68.51    | 73.12    | 60.32    | 63.01    | 82.60    | 90.43    |
> | 0-shot LIFT(GPT-3.5)              | 78.23  |  80.69   |  83.92   |  73.60   |  72.57   |  73.12   |  60.32   |  70.92   |  87.93   |  90.43   |
> | 0-shot TabLLM(GPT-3.5)            | 76.37  |  78.87   |  87.06   |  78.24   |  74.39   |  75.69   |  61.78   |  68.48   |  85.78   |  89.11   |
> | SERSAL(GPT-3.5, 0-shot prompting) | 91.39  |  85.42   |  86.40   |  79.39   |  85.14   |  78.97   |  63.97   |  76.36   |  96.85   |  98.37   |
> | SERSAL(GPT-3.5, 0-shot LIFT)      | 93.82  |  85.42   |  85.76   |  78.96   |  88.13   |  78.97   |  64.28   |  83.06   |  97.93   |  97.89   |
> | SERSAL(GPT-3.5, 0-shot TabLLM)    | 93.82  |  85.42   |  88.39   |  80.71   |  89.27   |  82.54   |  65.02   |  81.74   |  97.51   |  98.16   |
>
> As can be clearly seen, though the direct LLM prediction is slightly affected by prompt engineering, the significant bonus of applying SERSAL is still holds, which is more profound when we combine SERSAL with TabLLM or LIFT, even though they are primarily for LLM supervised fine-tuning. The results show the orthogonal technical contribution and novelty of SERSAL. We will sort the results and analysis into Appendix of the paper to specify the consideration of baseline settings and the technical inclusivity of SERSAL.
>
> **Reference:**
>
> [1] TabLLM: Few-shot Classification of Tabular Data with Large Language Models, AISTATS 2023.
>
> [2] LIFT: Language-Interfaced Fine-Tuning for Non-Language Machine Learning Tasks, NeurIPS 2022.
>
> ---
>
> **W4:** It seems the datasets evaluated in Table 2 are publicly available with most of them having been released for a long time, which produces a potential risk of being part of pre-training corpus of the used LLMs (i.e., GPT-3.5 here). Can we carry the evaluations on (i) private datasets; (ii) public datasets which are released after pre-training-data cutoff dates for the LLMs? Such evaluations on private or future data can help us figure out whether the SERSAL can help an old LLM produce high-quality confidence scores if we test on the tabular datasets in the future and real world.
>
> **A4:** Sincerely appreciate your thoughtful questions and suggestions! We agree that it is important and more reasonable to evaluate SERSAL on the very recent datasets to avoid their potential exposure to pre-training corpus of the used LLM, i.e., GPT-3.5 in this paper, which helps demonstrate the stronger generalizability of experiment findings of our SERSAL. **We would like to answer your questions from four facets**.

---

> ### Author Response · Authors · 2024-11-26
> **To m9i1 (5/6, Weakness 4)**
>
> (a) Actually, we have considered the mentioned potential risk of evaluation data exposure in the paper, and specially arrange evaluation on difficult tabular datasets of clinical trial tasks used in [1] (see the subsection “Continuous Performance Growth” in Sec. 3.2, and Table 7, datasets “N00041119”, “N00174655”, “N00312208”, “N00079274”, “N00694382”, we do not pose them in the main paper for the page limit), which are about mouse mortality prediction based on the new drug trial data. These clinical trial datasets are updated regularly and can only be accessed by submitting a real-name application on [NIH official website](https://clinicaltrials.gov/) for several-week approval procedure, and these datasets cannot be accessed by website crawlers. Also, our evaluated version is just released in 2024, which date is obviously later than the used GPT-3.5 pre-training corpus.
>
> (b) We additionally conduct SERSAL evaluations on even more older LLMs, online GPT-3 (175B, released at 2020 Mar, 45 TB corpus) & offline pre-trained GPT-2 (1.5B, released at 2019 Feb, 40 GB Web corpus) here, using datasets released after 2020 on Kaggle: LC (2021), ECD (2020, from real-world hospital in Sylhet, Bangladesh), ST (2020), CO (2020 May), AN (2021) from Table 2.
>
> | Dataset (release time):           | LC (2021) | ECD (2020 from hospital) | ST (2020) | CO (2020 May) | AN (2021) |
> |-----------------------------------|:---------:|:------------------------:|:---------:|:-------------:|:---------:|
> | Supervised FT-Transformer         |   86.61   |          99.60           |   82.98   |     99.91     |   99.92   |
> | 0-shot prompting(GPT-3.5)         |  78.87    |         85.71            |  63.01    |    82.60      |  90.43    |
> | SERSAL(GPT-3.5, 0-shot prompting) |   85.42   |          86.40           |   76.36   |     96.85     |   98.37   |
> | 0-shot prompting(GPT-3, 2020 Mar) |   73.58   |          82.30           |   60.80   |     75.36     |   88.62   |
> | SERSAL(GPT-3, 0-shot prompting)   |   82.06   |          85.17           |   71.69   |     88.13     |   95.22   |
> | 0-shot prompting(GPT-2, 2019)     |   63.76   |          68.53           |   54.28   |     63.28     |   73.16   |
> | SERSAL(GPT-2, 0-shot prompting)   |   71.24   |          73.28           |   57.93   |     70.52     |   80.68   |
>
> As expected, 0-shot prompting results of GPT-3 are slightly worse and GPT-2 are significantly worse than the ones of GPT-3.5, but our SERSAL bonus still holds, showing the SERSAL’s wide generalizability on both new or old LLMs.
>
> (c) We can also prove by contradiction to show your concern on exposure of our evaluated tabular data may be unnecessary. Here we compare the performance of original GPT-3.5 and GPT-3.5 fine-tuned on labeled training datasets (using simple prompt template as corpus, fine-tuned on three datasets: HF, ECD, HE) using 6 datasets: HF, LC, ECD, LI, HE, PID from Table 2 for evaluations.
>
> | Dataset:                                        |   HF   |    LC    |    ECD   |    LI    |    HE    |    PID   |
> |-------------------------------------------------|:------:|:--------:|:--------:|:--------:|:--------:|:--------:|
> | supervised FT-Transformer                       | 88.19  |  86.61   |  99.60   |  78.94   |  100.00  |  84.72   |
> | 0-shot prompting(GPT-3.5, original)             | 71.88  | 78.87    | 85.71    | 76.81    | 68.51    | 73.12    |
> | 0-shot prompting(GPT-3.5, trained on HF,ECD,HE) | 93.36  |  78.87   |  95.39   |  76.37   |  93.20   |  72.85   |
> | SERSAL(GPT-3.5 original, 0-shot prompting)      | 91.39  |  85.42   |  86.40   |  79.39   |  85.14   |  78.97   |
> | SERSAL(GPT-3.5 trained, 0-shot prompting)       | 94.18  |  85.42   |  95.39   |  79.39   |  95.38   |  79.36   |
>
> There are three apparent observations: (1) If GPT-3.5 is fine-tuned on one tabular datasets, its 0-shot prompting performance will be close to the supervised small model (FT-Transforner), which is significantly better than the result in our paper with a large margin; (2) The bonus of applying SERSAL to supervisedly trained GPT-3.5 on the exposed datasets (i.e., HF, ECD, HE) is limited, while in the paper the bonus is very strong, it is also contradictory that the performance gap between 0-shot prompting and SERSAL is so large if the base LLM has been pre-trained on the test dataset; (3) We find training on three medical tabular datasets does not impact the 0-shot prompting performance on the other three datasets, since the targets of our used medical datasets in Table 2 still differ in both features and diseases, increasing the difficulty of sharing potential patterns from one vertical disease to another one.
>
> **Reference:**
>
> [1] TransTab: Learning Transferable Tabular Transformers Across Tables, NeurIPS 2022.

---

> ### Author Response · Authors · 2024-11-26
> **To m9i1 (6/6, Weakness 4&5)**
>
> (d) In real-world practice, the users often choose the current SOTA LLMs for various downstream tasks, since the low-shot generalizability of powerful LLMs is from its wide enough pre-training corpus. And the major target of our SERSAL is to help the LLMs perform better inference on the tabular data, compared to only using prevailing textual prompting techniques, which advantage remains unchanged regardless of whether the tabular data has been pre-trained or not.
>
> ---
>
> **W5:** Why only the results of ECD and LI datasets shown in Fig. 2 and Table 4, what about the results of other datasets?
>
> **A5: We have appended the illustration of performances in different LLM confidence ranges on other datasets in Fig. 4 of the Appendix, and additionally conduct multi-loop SERSAL on other datasets with results added to the Table 8 of the Appendix**. The reason we originally just used ECD and LI datasets in Table 4 is that: (1) The performances of these two datasets are close to the ones of baselines, thus we choose them for multi-loop evaluation to inspect the further potential of SERSAL; (2) To save the cost of ChatGPT API; (3) Paper page limit. The additional results in Table 8 are as follows:
>
> |                                            |   HF   |   LC   |   ECD  |   LI   |   HE   |   PID  |   FH   |   ST   |   CO   |   AN   |
> |--------------------------------------------|:------:|:------:|:------:|:------:|:------:|:------:|:------:|:------:|:------:|:------:|
> | 0-shot GPT-3.5 @ loop#1 (original GPT-3.5) | 71.88  | 78.87  | 85.71  | 76.81  | 68.51  | 73.12  | 60.32  | 63.01  | 82.60  | 90.43  |
> | SERSAL @ loop#1                            | 91.39  | 85.42  | 86.40  | 79.39  | 85.14  | 78.97  | 63.97  | 76.36  | 96.85  | 98.37  |
> | 0-shot GPT-3.5 @ loop#2                    | 87.58  | 83.74  | 86.42  | 80.26  | 86.18  | 79.26  | 63.86  | 73.62  | 91.29  | 93.62  |
> | SERSAL @ loop#2                            | 92.03  | 86.15  | 87.00  | 82.47  | 87.32  | 80.61  | 65.27  | 79.58  | 97.20  | 98.93  |
> | 0-shot GPT-3.5 @ loop#3                    | 89.26  | 85.39  | 87.81  | 82.91  | 86.87  | 81.47  | 64.12  | 76.37  | 93.65  | 94.13  |
> | SERSAL @ loop#3                            | 93.58  | 85.42  | 89.00  | 84.07  | 89.57  | 81.83  | 65.27  | 80.93  | 97.02  | 98.60  |
>
> As we can see, the bonus of continuously performing SERSAL also widely exists in other datasets, while the most performance gain has been acquired at the first-loop SERSAL process, showing the significance of learning from the LLM annotations to refine the inherent knowledge at the first time, and further improvement is minor and tends to gradually diminish.
>
> For the extra illustrations in Fig. 4 of Appendix, the overall trend is that, though the performances on the samples with low LLM confidence are fluctuating across different datasets, which is impacted by the base LLM itself, the samples in high LLM confidence are stably reliable for early stopping. Notably, in SERSAL process clean samples are not equivalent to the samples in high LLM confidence, but the ones easier to fit by the small tabular prediction model, which is judged by the DivideMix algorithm, while high-confidence samples are more likely to be clean during DivideMix.
>
> ---
>
> Thank you for your different voices and understanding, we hope our response can address your major concerns and confusions. If you have any further questions or concerns, feel free to raise them and discuss. Thank you very much.

---

> > ### Comment · Reviewer_m9i1 · 2024-11-30
> > **Response to Authors**
> >
> > **Response to A1:** Thank you for the detailed clarifications. While I agree that the resulting capabilities of SERSAL can in some ways be viewed as “extending” the capabilities of LLMs, I still do not think that this is an approach that improves the “zero-shot” generalizability of LLMs for tabular prediction, as you are explicitly fine-tuning the LLM and the the downstream prediction model using unlabeled test data. Both “zero-shot tabular prediction” and “zero-shot learning” seem to be misnomers. It is true that for this approach to work properly, the initial zero-shot prompting predictions from the LLMs have to be reasonable and better than random, but the alternating refinement procedure involves fine-tuning on unlabeled data.
> >
> > **Response to A2:** Great, I think adding these details would significantly improve the clarity of the algorithm description.
> >
> > **Response to A3:** Thank you for clarifying. I would suggest making the points in A3-2 and A3-3 more explicit and upfront for better emphasis on the contributions of the proposed work.
> >
> > **Response to A4:** I do not fully agree with your discussion in (c), but I also understand that this is in general a tricky problem to address, and I appreciate the additional effort. Moreover, I think the additional results and discussions in (a) and (b) are interesting and help address some of these memorization concerns, and I think it would be good to discuss these results in the paper (even if it is deferred to the Appendix). For example, for (a), it doesn’t seem like these details are explicitly discussed.
> >
> > **Response to A5:** Fig. 4 in the Appendix seems to suggest that the LLM confidence scores can be significantly uncalibrated, much more so than the ECD and LI datasets shown in Fig. 2. I think the fact that the LLM confidence scores can be unstable and therefore affect early stopping behavior should be noted as a limitation/caveat for the proposed work.
> >
> > Overall, I appreciate the additional clarifications provided by the authors. In light of the updates and given the good empirical performance demonstrated in the paper, I have raised my score to a 5.

---

> > > ### Author Response · Authors · 2024-12-03
> > > **Sincere thanks for the point-by-point feedback and improved value recognition**
> > >
> > > We sincerely appreciate your detailed feedback to express the attitude change on our work after rebuttal. Although it seems there existed slight misalignment on research value or smell between us, we respect different voices and believe such collision of ideas is beneficial for promoting academic development. We are very glad to see that we improve your recognition from the perspective of strong empirical performance through our effort on additional demonstration and experiment to robustly hold the effectiveness of SERSAL method. We believe the SERSAL can be a practical tool to unconditionally improve LLM performance on tabular prediction tasks and its technical landscape can be a promising research direction to reveal the untapped potential of LLMs in the future.

---

### Meta-Review · Area_Chair_zEe6 · 2024-12-17

**Metareview:**

This paper presents a novel approach to unsupervised tabular learning by combining large language models (LLMs) with traditional tabular data prediction methods. These two models are trained iteratively using unlabeled datasets to enhance their performance. The experimental results demonstrate consistent improvements over alternative LLM-based tabular learning methods across various medical datasets.

The proposed method offers a novel and intriguing approach, effectively leveraging the synergy between traditional tabular data prediction models and LLMs. The extensive experiments are promising, and the authors further demonstrate the generalizability of their approach by providing additional results from non-medical settings during the rebuttal, which have been incorporated into the revised manuscript.

However, I concur with other reviewers that the claim of a "zero-shot" setting is inaccurate.  The authors should clarify this as an "unsupervised" setting and adjust the paper's positioning accordingly.

Despite this weakness, the strengths of this paper outweigh its limitations. The proposed method is innovative and well-supported by empirical evidence. The issue of terminology can be easily addressed in the camera-ready version. Therefore, I recommend accepting this paper.

**Additional Comments On Reviewer Discussion:**

The authors did a great job addressing the reviewers' concerns during the rebuttal period. They significantly strengthened the paper by providing additional results across different domains and tasks, including those beyond binary prediction, which greatly enhances the proposed method's applicability.

While the authors haven't yet fully addressed the comments regarding the "zero-shot" and "unsupervised" settings, this can be rectified in the camera-ready version. I agree with the reviewers that clarifying the distinction between these settings and accurately reflecting the unsupervised nature of the work is crucial. I encourage the authors to carefully consider these valid points and revise the manuscript accordingly.

---

### Decision · Program_Chairs · 2025-01-22

Accept (Poster)